# `FOOGD`: Federated Collaboration for Both Out-of-distribution Generalization and Detection

**Xinting Liao[1], Weiming Liu[1], Pengyang Zhou[1], Fengyuan Yu[1], Jiahe Xu[1],**

**Jun Wang[2], Wenjie Wang[3], Chaochao Chen[1], Xiaolin Zheng[1]** [*]

[1] Zhejiang University, [2] OPPO Research Institute, [3] National University of Singapore
{xintingliao, 21831010, zhoupy, zjuccc, xlzheng}@zju.edu.cn,
junwang.lu@gmail.com, wenjiewang96@gmail.com

## Abstract

Federated learning (FL) is a promising machine learning paradigm that collaborates with client models to capture global knowledge. However, deploying FL models in real-world scenarios remains unreliable due to the coexistence of in-distribution data and unexpected out-of-distribution (OOD) data, such as covariate-shift and semantic-shift data. Current FL researches typically address either covariate-shift data through OOD generalization or semantic-shift data via OOD detection, overlooking the simultaneous occurrence of various OOD shifts. In this work, we propose `FOOGD`, a method that estimates the probability density of each client and obtains reliable global distribution as guidance for the subsequent FL process. Firstly, `SM`$^3$`D` in `FOOGD` estimates score model for arbitrary distributions without prior constraints, and detects semantic-shift data powerfully. Then `SAG` in `FOOGD` provides invariant yet diverse knowledge for both local covariate-shift generalization and client performance generalization. In empirical validations, `FOOGD` significantly enjoys three main advantages: (1) reliably estimating nonnormalized decentralized distributions, (2) detecting semantic shift data via score values, and (3) generalizing to covariate-shift data by regularizing feature extractor. The preject is open in `https://github.com/XeniaLLL/FOOGD-main.git`.

## 1 Introduction

Federated learning (FL) [56] provides a distributed machine learning paradigm, which collaboratively models decentralized data resources. Specifically, each client models its data locally and server improves model performance by aggregating client models, which indirectly shares knowledge among clients and preserves privacy. FL further makes efforts to adapt real-world scenarios, i.e., adapting non-independent and identical distribution (*non-IID*) [39, 30].

Beyond non-IID issues, deploying FL models in real-world also encounters different tasks of out-of-distribution (OOD) shift [69, 26, 6], e.g., tackling covariate shifts (*OOD generalization*) and handling semantic shifts (*OOD detection*). In FL, OOD generalization task is devised to capture the invariant data-label relationships of covariate-shift data intra- and inter-client, which offers the potential of adapting unseen clients [17, 60, 70, 84]. The OOD detection task in FL aims to find semantic-shift data samples that do not belong to any known categories of all client data during FL training [83]. Both OOD generalization and detection simultaneously exist in FL, hindering the deployment of FL methods. Nevertheless, the existing work only tackles each OOD task in isolation. SCONE [3] proposes a unified margin-based framework to realize OOD generalization and OOD detection tasks

---

[*]Corresponding author.

in centralized machine learning. But it is infeasible to FL due to two reasons, i.e., being non-trivial in searching for consistent margin among non-IID distribution, and requiring outlier exposure of data. This motivates us to a crucial yet unexplored question:

*Can we devise a FL framework that adapts to wild data, which coexists with non-IID in-distribution (IN) data, covariate-shift (IN-C) data, and semantic-shift (OUT) data?*

In this work, we simultaneously promote OOD generalization and detection by collaborating with clients in FL. The objectives of OOD generalization and detection vary among different clients due to their non-normalized and heterogeneous probability densities. This motivates us to build systematic and global guidance to distinguish IN, IN-C, and OUT data. As depicted in Fig. 1, for non-IID client distributions, we first estimate the probability density in each local client and then compose these local estimations for global distribution in server. Once a reliable global distribution estimation is established, we

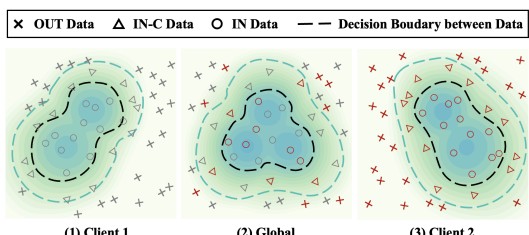

Figure 1: Motivation of FOOGD. The distributions of two clients are non-IID, and we seek to estimate the global distribution among decentralized data.

can leverage it to guide FL OOD tasks in deployment. However, this approach presents two challenges, i.e., *CH1: How to estimate the reliable and global probability density among decentralized clients for detection?* and *CH2: How to enhance intra- and inter-client OOD generalization based on global distribution estimation?*

To fill these gaps, we propose a federated collaboration framework named as FOOGD, which estimates client distribution in feature space via score matching with maximum mean discrepancy ($SM^3D$) and enhances the client model generalization by Stein augmented generalization (SAG). **To solve CH1**, inspired by the flexibility of score matching [58, 7], we originally devise $SM^3D$ to train score model that estimates limited and heterogeneous data distributions for each client, and aggregate score models in server as global estimation. Because the score values are vectors indicating position and changing degree of the log data density [55], $SM^3D$ brings the potential of discriminating OUT data in low-density areas with large change degree. However, it is unreliable to directly apply vanilla score matching for modeling decentralized data, which suffers from sparsity and multi-modal complexity [72, 55]. To obtain a reliable density estimation, $SM^3D$ explores wider space by generating random samples via Langevin dynamic sampling, and constrains the generated samples to be similar to data samples via maximum mean discrepancy (MMD). **To mitigate CH2**, SAG regularizes feature invariance between data samples and its augmented version, which is measured by score-based discrepancy. Though the existing generalization methods capture the invariance in feature space [1, 34], the vital feature information is inevitably lost due to strictly invariant constraints [87, 10]. This also deteriorates the performance of solving FL OOD generalization. With the benefits of distributional alignment based on Stein indentity [46], SAG in client model captures IN-C data in a similar feature space with IN data, which not only avoids representation collapse but also maintains diversifying information. Thus SAG makes FOOGD generalize to IN-C data from local covariate-shift distribution and unseen client distribution.

The main contributions are: (1) We are the first to study OOD generalization and detection in FL simultaneously, and formulate a evaluation on deploying FL methods in the wild data. (2) We propose FOOGD which estimates reliable global distributions based on arbitrary client probability densities, to guide both OOD generalization and detection. (3) We devise $SM^3D$ which not only explores wider probability space for density estimation, but also provides the score function values to detect OUT samples. (4) We utilize SAG to maintain the invariance between IN-C and IN data in feature space, which obtains better generalization without collapsing for FL scenarios. (5) We provide theoretical analyses and conduct extensive experiments to validate the effectiveness of FOOGD.

## 2 Related Works

### 2.1 OOD Detection

OOD detection discriminates semantic shift (OUT) data during deployment time [3, 20, 53]. There are two main categories of OOD detection work, i.e., enhancing training-time regularization [48,

21, 75, 13], and measuring post-hoc detection function of a well-trained model [20, 74, 37]. The first category focuses on ensuring predictors produce low-confidence predictions for OOD data during training, which is effective but mainly requires access to real OUT data [21, 5, 80, 86]. By the way, selecting different auxiliary detection objectives [22, 57] unexpectedly varies the overall performance. The second category utilizes the classification logits [74], energy score [35, 48], and feature space estimation [37, 66] from pre-trained models, to detect the OUT data. This reduces the costly computation burden but rigidly relies on the data distribution captured in the pre-trained model. As one kind of methods in post-hoc way, density-based estimation methods [37, 66, 68] can relieve the cost of collecting or synthesizing representative OOD datasets, avoiding biased and ineffective detection [74, 35] and bringing the potential of densities composition.

## 2.2 OOD Generalization

OOD generalization targets extracting invariant feature-label relationships and maintaining the deployment performance of model with covariate-shift data in the open-world [31, 54]. To reach this goal, IRM-based work [2, 1, 34] utilizes invariant risk regularization to find invariant representations from different covariate shift data. Besides, there are various work calibrating invariant representations by distribution robust optimization methods [65, 16], feature alignment methods [15, 14], augmentaed training [10], gradient manipulation methods [24], diffusion modeling [82] and so on. SCONE [3] takes advantage of unlabeled wild mixture data to enhance generalization and build detectors simultaneously. However, SCONE is not suitable for FL, since it requires a hyper-parameter of energy margin and the outlier exposure data [83, 75]. To tackle the meta-task detection and generalization, Chen[9] propose an Energy-Based Meta-Learning (EBML) framework that learns meta-training distribution via two energy-based neural networks. However, it is tough to model two reliable energy models in decentralized models where data and computation resources are constrained.

## 2.3 Federated Learning with Wild Data

In FL, wild data makes it challenging in tackling non-IID modeling, OOD generalization, and OOD detection. Firstly, FL with non-IID data presents significant challenges in balancing global and local model performance [56, 8, 39, 43, 89, 42](Appendix C). Secondly, FL considers two aspects of generalization, i.e., (1) intra-client generality, and (2) inter-client generality. The intra-client generalization keeps the invariant relationship between data samples and class labels[26, 69, 70, 63], which is similar to centralized OOD generalization. The inter-client generality work captures invariant representation for heterogeneous client distributions, making the global model adaptive to a newly unseen client [84, 60, 17, 44]. Lastly, regarding OOD detection in FL, it is expected to detect semantic shift data out of the whole class categories set among decentralized data, yet avoid wrongly distinguishing unseen data classes of other clients. FOSTER [83] treats unseen data classes in each client as OUT, and enhances their detection capability via synthesizing virtual data with external classes of other clients. Different from the above methods, we aim to enhance OOD detection and generalization simultaneously by collaborating with different clients. Recently, FedGMM [77] utilizes a federated expectation-maximization algorithm to fit data distribution among clients by estimating Gaussian mixture models(GMM), and detects OUT data via computing GMM probability. It can only roughly capture the data distribution with the prior assumption of GMM. Meanwhile, a orthogonal paradigm of studies focus on tackling concept shifts in federated process [61, 29]. However, it overlooks the coexistence of wild data, resulting in suboptimal performance in federated tasks of OOD generalization and detection.

# 3 Methodology

## 3.1 Problem Setting

**Federated Learning Formulation with Wild Data.** We first formulate the wild data in FL deployment and provide the optimization goal of FL. Empirically, we assume a dataset decentralizes among $K$ clients, i.e., $\mathcal{D} = \cup_{k \in [K]} \mathcal{D}_k$. The data distribution of $k-$th client is simulated following the real-world wild data, i.e., $\mathcal{D}_k = \mathcal{D}_k^{\text{IN}} + \mathcal{D}_k^{\text{IN-C}} + \mathcal{D}_k^{\text{OUT}}$. The objective of the FL model, which simultaneously tackles OOD generalization and detection, is defined as follows:

$$\text{argmin}_{\boldsymbol{\theta}_f, \boldsymbol{\theta}_g} \Sigma_{k=1}^K w_k \mathbb{E}_{\boldsymbol{x} \sim p_{\mathcal{D}_k}} [\mathcal{L}_k(\boldsymbol{\theta}_f, \boldsymbol{\theta}_g; \mathcal{D}_k)], \tag{1}$$

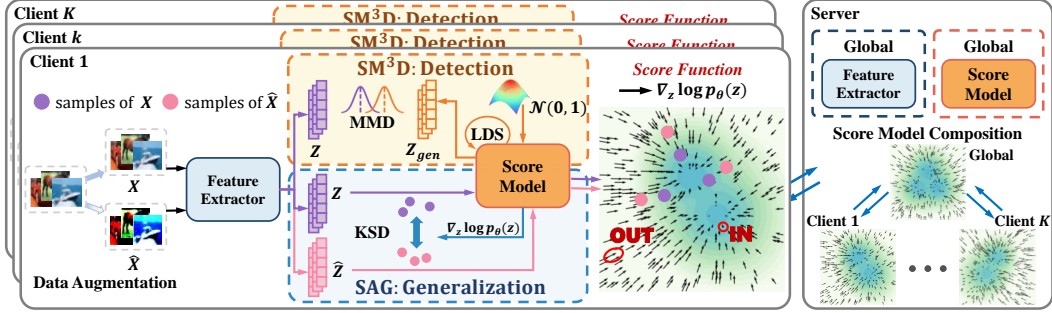

Figure 2: Framework of FOOGD. For each client, we have main task feature extractor, a SM$^3$D module estimates score model (Eq. (8)) for detection, and a SAG module regularizes feature extractor for enhancing generalization. The server aggregates models and obtains global distribution.

where $\mathcal{L}_k(\boldsymbol{\theta}_g, \boldsymbol{\theta}_f; \mathcal{D}_k) = \ell_k^{\text{IN}} + \ell_k^{\text{IN-C}} + \ell_k^{\text{OUT}}$, and $w_k$ represents weight ratio for the $k$-th client. The OOD measurements $\ell_k^{\text{IN}}, \ell_k^{\text{IN-C}}, \ell_k^{\text{OUT}}$ correspondingly justify the IN generalization, IN-C generalization, and OUT detection in each client $k$, as follows:

$$\ell_k^{\text{IN}} := \mathbb{E}_{(\boldsymbol{x},y) \sim P_{\mathcal{D}_k^{\text{IN}}}} \left( \mathbb{I} \{ y_{\text{pred}} \left( f_{\boldsymbol{\theta}}(\boldsymbol{x}) \right) \neq y \} \right) \qquad (a)$$

$$\ell_k^{\text{IN-C}} := \mathbb{E}_{(\boldsymbol{x},y) \sim P_{\mathcal{D}_k^{\text{IN-C}}}} \left( \mathbb{I} \{ y_{\text{pred}} \left( f_{\boldsymbol{\theta}}(\boldsymbol{x}) \right) \neq y \} \right) \qquad (b) \qquad (2)$$

$$\ell_k^{\text{OUT}} := \mathbb{E}_{(\boldsymbol{x},y) \sim P_{\mathcal{D}_k^{\text{OUT}}}} \left( \mathbb{I} \{ g_{\boldsymbol{\theta}}(\boldsymbol{x}) = \text{IN} \} \right) \qquad (c),$$

where $f_{\boldsymbol{\theta}}(\cdot)$ is main task model, $g_{\boldsymbol{\theta}}(\cdot)$ is detector, $\mathbb{I}$ is indicator function, and $y_{\text{pred}}$ is predicted label.

**Framework Overview.** To optimize the FL objective in Eq. (1), we propose FOOGD whose framework overview is depicted in Fig. 2. For $K$ clients with non-IID data, FOOGD composes their local distributions and aggregate their model parameters in server. In each client, the data samples $\boldsymbol{x}$ as well as its fourier augmented [79] counterparts $\widehat{\boldsymbol{x}}$, are fed into the same feature extractor of main task $f_{\boldsymbol{\theta}}(\cdot)$ to obtain their latent features, $\boldsymbol{z} = f_{\boldsymbol{\theta}}(\boldsymbol{x})$ and $\widehat{\boldsymbol{z}} = f_{\boldsymbol{\theta}}(\widehat{\boldsymbol{x}})$, respectively. To avoid overwhelming communication costs brought by score models, score matching with maximum mean discrepancy (SM$^3$D) trains a score model $s_{\boldsymbol{\theta}}(\cdot)$ in feature space. This model captures the data distribution by estimating the gradient of log densities (score functions) of latent features $\boldsymbol{z}$, i.e., $s_{\boldsymbol{\theta}^*}(\boldsymbol{z}) = \nabla_{\boldsymbol{z}} \log p_{\boldsymbol{\theta}}(\boldsymbol{z}) \approx \nabla_{\boldsymbol{z}} \log p_{\mathcal{D}}(\boldsymbol{z})$ [7, 72]. Then score model serves as the detector for the objective in Eq. (2c), discriminating OUT based on the norm of score function values. Besides, Stein augmented generalization (SAG) enhances the generalization capabilities of the feature extractor $f_{\boldsymbol{\theta}}(\cdot)$, by the distribution regularization defined via score model. Because score model based distribution ensures that data features and their neighboring augmented samples, e.g., $\boldsymbol{z}$ and $\widehat{\boldsymbol{z}}$, maintain a consistent probability space [46]. The local modeling iterates until performance converges.

In each communication round, since both main task model and score model are parameterized neural networks, it is practical to follow conventional weighted average aggregation [56], i.e.,

$$\{\boldsymbol{\theta}_s, \boldsymbol{\theta}_f\} = \sum_{k=1}^{K} w_k \{\boldsymbol{\theta}_s^k, \boldsymbol{\theta}_f^k\}, \qquad (3)$$

with $w_k = \frac{|\mathcal{D}_k|}{\sum_{k=1}^{K} |\mathcal{D}_k|}, \forall_{k \in [K]}$. These collaborative processes among clients continue until the global model converges, bringing reliable and comprehensive global distribution in the form of global score model. We introduce the details later and illustrate the algorithm of FOOGD in Appendix A Algo. 1.

### 3.2 SM$^3$D: Estimating Score Model for Detection

In this part, we introduce the estimation of FL data distribution and how to utilize it for detection. As shown in Fig. 1, a reliable probability density is eagerly necessary for distinguishing IN and OUT data [75, 27]. Different from existing centralized OUT aware and OUT synthesis methods [13, 21], the FL framework suffers from the accessibility of OUT data [83]. In this study, we aim to explicitly capture the local IN data distribution of clients, and subsequently compose them to reliable global

distribution for discrimination. However, it remains challenging to estimate heterogeneous and non-normalized probability density without prior information during FL modeling.

**Dynamic Feature Density Estimation.** FOOGD estimates score model via score matching in the feature space [72, 32, 7], i.e., $p_\mathcal{D}(z)$, circumventing the need for prior distribution knowledge or distribution normalization [72]. Moreover, it alleviates the computational burden by modeling the score of latent representations in a smaller, yet more expressive and continuous space, compared to the scores of the original data [71]. Specifically, given the latent features $z = f_\theta(x)$, we perturb it via adding random noise $v \sim \mathcal{N}(\mathbf{0}, \mathbf{I})$ to obtain $\tilde{z} = z + \sigma v$, which follows noise-perturbed data distribution $p_\sigma(\tilde{z}|z) := \mathcal{N}\left(\tilde{z}; z, \sigma^2 \mathbf{I}\right)$. And we model it with noise conditional score model [67] $s_\theta(\tilde{z}, \sigma)$ by minimizing the denoising score matching (DSM) loss, i.e.,

$$\min \ell_{\text{DSM}} = \frac{1}{2}\mathbb{E}_{p_\mathcal{D}(z)p_\sigma(\tilde{z}|z)} \left\| s_\theta(\tilde{z}, \sigma) - \nabla_{\tilde{z}} \log p_\sigma(\tilde{z} \mid z) \right\|^2, \tag{4}$$

where the score function of $\nabla_{\tilde{z}} \log p_\sigma(\tilde{z} \mid z)$ for $d$-dimensional features, is computed as follows:

$$\nabla_{\tilde{z}} \log p(\tilde{z} \mid z) = \nabla_{\tilde{z}} \left[ \log \frac{1}{\left(\sqrt{2\pi\sigma^2}\right)^d} \exp\left\{ -\frac{\|\tilde{z} - z\|^2}{2\sigma^2} \right\} \right] = -\frac{\tilde{z} - z}{\sigma^2} = -\frac{v}{\sigma}. \tag{5}$$

When the noise get to zero, i.e., $\sigma \to 0$, we have the exact score values $s_\theta(\tilde{z}, \sigma) = s_\theta(z)$. However, score model based density estimation will inevitably fail once the distribution contains sparse data samples [67, 55, 67] or multiple modalities [32], as shown in Fig. 3 (a). SM³D is motivated to broadly explore the generated random features $z_{\text{gen}}$ that samples from the whole distribution space. In detail, SM³D first sample from a random distribution, e.g., Normal distribution, as the start latent features, i.e., $z^0 \sim \mathcal{N}(\mathbf{0}, \mathbf{I})$. Then SM³D utilizes $T$-step Langevin dynamic sampling [67] (LDS) from density vector fields modeled by the score model, to derive generated latent features $z_{\text{gen}} = z^T$:

$$z^t = z^{t-1} + \frac{\epsilon}{2} s_\theta(z^{t-1}, \sigma) + \sqrt{\epsilon} w^t, \tag{6}$$

with $\epsilon$ indicating the step size and $w^t \sim \mathcal{N}(\mathbf{0}, \mathbf{I})$ introducing stochasticity in each step. Lastly, the distribution of a batch of the generated features $Z_{\text{gen}} = \{z_{\text{gen},i}\}_{i=1}^B$, i.e., $p_{\text{gen}}(z_{\text{gen}})$, is supposed to approximate the distribution of original features $Z = \{z_i\}_{i=1}^B$, i.e., $p_\mathcal{D}(z)$, with the calibration of maximum mean discrepancy ($\text{MMD}(Z, Z_{\text{gen}})$) matching:

$$\ell_{\text{MMD}} = \mathbb{E}_{z_\mathcal{D}, z'_\mathcal{D} \sim p_\mathcal{D}}[k(z_\mathcal{D}, z'_\mathcal{D})] - 2\mathbb{E}_{z_\mathcal{D} \sim p_\mathcal{D}, z'_{\text{gen}} \sim p_{\text{gen}}}[k(z_\mathcal{D}, z'_{\text{gen}})] + \mathbb{E}_{z_{\text{gen}}, z'_{\text{gen}} \sim p_{\text{gen}}}[k(z_{\text{gen}}, z'_{\text{gen}})]. \tag{7}$$

where $k(z, z') = \exp(\frac{1}{h}\|z - z'\|^2)$ with bandwidth $h$ is Gaussian kernel function [47, 49] within a unit ball in universal Reproducing Kernel Hilbert Space (RKHS). Because MMD is a non-parametric method that accurately measures the distance between two densities in RKHS, it provides reliable estimations and adapts well to complex data modalities [4]. This approach mitigates the limitations of directly using DSM to estimate distributions by exploring a wider feature space. Unfortunately, as depicted in Fig. 3 (d), simply using MMD matching does not enhance density estimation, when the target distribution is unknown or inaccurate. But it is quite necessary that the latent distribution is inaccurate and heterogeneous in FL. To fill this gap, SM³D seeks to harness and integrate the strengths of both density estimation paradigms, via a trade-off coefficient $\lambda_m$:

$$\ell^{\text{OUT}} = (1 - \lambda_m)\ell_{\text{DSM}} + \lambda_m \ell_{\text{MMD}}. \tag{8}$$

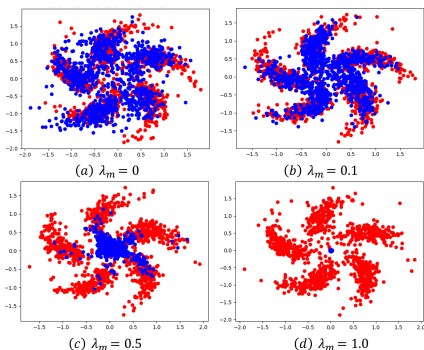

*(a)* $\lambda_m = 0$   *(b)* $\lambda_m = 0.1$

*(c)* $\lambda_m = 0.5$   *(d)* $\lambda_m = 1.0$

Figure 3: Motivation of SM³D. Red points are sampled from target data distribution, and the blue points are generated by LDS in Eq. (6).

In this way, SM³D brings an accurate and flexible implementation for non-normalized data distribution. The implementation procedure of SM³D is in Appendix A Algo.2. To illustrate the effectiveness of SM³D, we further visualize a density estimation of 2-D toy example in Fig. 3. In detail, we model the red target points by tuning a series of coefficients, i.e., $\lambda_m = \{0, 0.1, 0.5, 1\}$ in Eq. (8). As we

can see, with the mutual impacts between score matching and MMD estimation, SM$^3$D has more compact density estimation when $\lambda_m = 0.1$, compared with blankly using score matching ($\lambda_m = 0$) or simply using MMD ($\lambda_m = 1$). As a brand new objective of density estimation, SM$^3$D expands the searching range and depth of score modeling, making it possible to comprehensively model data density. Moreover, with the calibration of MMD estimation, original data features and the generated samples based on the score model are effectively matched. Hence SM$^3$D could ensure a more aligned and reliable density estimation for sparse and multi-modal data.

**OOD detection in clients.** Remind that the score function indicates the gradient of the log density, which are actually vector fields pointing to the highest density area, as shown in the score function visualization of Fig. 2. The IN data should point to the high density and reflect the distance via its vector norm. While the OUT data cannot present this satisfying property and further exposure boldly, since the OUT data is always in low-density area [48, 55]. That is, the norm of the score $\|s_{\boldsymbol{\theta}^*}(\boldsymbol{z})\| = \|\frac{\nabla_z p_{\boldsymbol{\theta}^*}(\boldsymbol{z})}{p_{\boldsymbol{\theta}^*}(\boldsymbol{z})}\|$ decreases in regions of higher density, while increases in lower density. It indicates the larger the norm of score is, the more likely the data sample is OUT. For *negative threshold* $\tau < 0$, we have detection score function:

$$\text{IsOUT}(\boldsymbol{x}) = \text{True}, \quad \text{when} \quad \|s_{\boldsymbol{\theta}^*}(f_{\boldsymbol{\theta}^*}(\boldsymbol{x}))\| > -\tau; \quad \text{otherwise}, \quad \text{IsOUT}(\boldsymbol{x}) = \text{False}. \quad (9)$$

### 3.3 SAG: Enhancing Feature Extractor for Gerneralization

In this section, we will illustrate how to enhance generalization capability of feature extractor in FOOGD. In FL scenarios, solving OOD generalization needs not only to keep the local IN-C data classification correctly, but also to maintain performance consistency among all participating clients. The non-IID issue creates a contradiction between achieving both targets. This is because enhancing IN-C accuracy intra-client requires diversification across different classes, whereas inter-client generalization benefits from all IN data being closely clustered, irrespective of class distinctions. Hence, it is expected to balance the feature diversification of different classes and the feature consistency of in-distribution, to realize the consistent data-label relationships intra- and inter-client.

**Diversifying Feature Invariance Augmentation.** FOOGD regularizes invariance among client feature extractors using distribution-aware divergence between the original data $\boldsymbol{x}$ and its augmented version $\widehat{\boldsymbol{x}} = \mathcal{T}(\boldsymbol{x})$ by transformation $\mathcal{T}$. To address this, we propose SAG, which utilizes global distribution and optimizes distributional invariance between latent features of the original and augmented data. This approach maintains the distinguishable diversification of features and consistent data-label mapping across clients.

In the feature space, SAG regularizes original data samples to be aligned with augmented ones, i.e., aligning $\boldsymbol{z} = f_{\boldsymbol{\theta}}(\boldsymbol{x})$ and $\widehat{\boldsymbol{z}} = f_{\boldsymbol{\theta}}(\widehat{\boldsymbol{x}})$. However, directly computing the norm regularization between $\boldsymbol{z}$ and $\widehat{\boldsymbol{z}}$ will cause mode collapse [28] in FL, further degrading the estimation of score model $s_{\boldsymbol{\theta}}(\cdot)$ based on SM$^3$D. While contrastive methods [62, 73, 51, 50] like FedICON [69], and L-DAWA [64] ensure diversification and alignment for generalization, they rely on selecting negative samples instead of leveraging global distribution knowledge. Consequently, they fail to maintain consistent invariance among clients. Instead, SAG alternatively introduces kernelized Stein operator guided by score function, i.e.,

$$\mathcal{A}_p \phi(\boldsymbol{z}) = \phi(\boldsymbol{z}) \nabla_{\boldsymbol{z}} \log p(\boldsymbol{z}) + \nabla_{\boldsymbol{z}} \phi(\boldsymbol{z}), \quad (10)$$

where $\phi(\boldsymbol{z})$ is implemented with kernel function $k(\cdot, \cdot)$ mentioned in Eq. (7) [46], while $p(\boldsymbol{z})$ and $q(\widehat{\boldsymbol{z}})$ are the distributions for a batch of features $\boldsymbol{Z} = \{\boldsymbol{z}_i\}_{i=1}^{B}$, and $\widehat{\boldsymbol{Z}} = \{\widehat{\boldsymbol{z}}_i\}_{i=1}^{B}$, respectively. By utilizing the kernelized Stein operator, SAG encourages the samples of augmented features to align with high probability regions of the original features. Additionally, the second term of (10) improves feature diversification and prevents data from collapsing directly to the original distribution modes. According to a fundamental theory named as Stein identity, i.e., $\mathbb{E}_{q(x)}[\mathcal{A}_q \phi(x)] = 0$ for arbitrary distribution $q(x)$ [46], Stein operator brings the potential of measuring two data distributions with the guidance of global distribution estimation. Because score models capture local probability densities and are aggregated into a global score model on the server, they inherit distribution information from all participating clients. Specifically, we first illustrate kernelized Stein discrepancy (KSD) [46, 41, 45] that measures the distribution discrepancy between original data $p(\boldsymbol{z})$ and augmented data $q(\widehat{\boldsymbol{z}})$:

$$\begin{aligned} \text{KSD}(p(\boldsymbol{z}), q(\widehat{\boldsymbol{z}})) = \mathbb{E}_{\widehat{\boldsymbol{z}}, \widehat{\boldsymbol{z}}' \sim q}[s_{\boldsymbol{\theta}}(\widehat{\boldsymbol{z}})^\top s_{\boldsymbol{\theta}}(\widehat{\boldsymbol{z}}') k(\widehat{\boldsymbol{z}}, \widehat{\boldsymbol{z}}') + s_{\boldsymbol{\theta}}(\widehat{\boldsymbol{z}})^\top \nabla_{\widehat{\boldsymbol{z}}'} k(\widehat{\boldsymbol{z}}, \widehat{\boldsymbol{z}}') \\ + s_{\boldsymbol{\theta}}(\widehat{\boldsymbol{z}}')^\top \nabla_{\widehat{\boldsymbol{z}}} k(\widehat{\boldsymbol{z}}, \widehat{\boldsymbol{z}}') + \text{trace}(\nabla_{\widehat{\boldsymbol{z}}} \nabla_{\widehat{\boldsymbol{z}}'} k(\widehat{\boldsymbol{z}}, \widehat{\boldsymbol{z}}'))]. \end{aligned} \quad (11)$$

We provide the full induction of KSD between original data and augmented data in Appendix B.2. And $\mathrm{KSD}(p(\boldsymbol{z}), q(\widehat{\boldsymbol{z}}))$ equals zero if and only if $p(\boldsymbol{z})$ and $q(\widehat{\boldsymbol{z}})$ are the same. By taking the derivative of KSD, we can obtain the updating direction of moving $\widehat{\boldsymbol{z}}$ towards $\boldsymbol{z}$, which not only keep the invariance of features, but also guarantee diversification avoiding collapse. Therefore, the augmented representation $\widehat{\boldsymbol{Z}}$ has minimal KSD with the original latents $\boldsymbol{Z}$. This ensures that the final objective of the feature extractor is to minimize the subsequent classification error (between predictions $\boldsymbol{Y}_{\mathrm{pred}}$ and ground truth $\boldsymbol{Y}_{\mathrm{gr}}$) and achieve invariant alignment:

$$\ell^{\mathrm{IN}} + \ell^{\mathrm{IN\text{-}C}} = \mathrm{CrossEntropy}(\boldsymbol{Y}_{\mathrm{pred}}, \boldsymbol{Y}_{\mathrm{gr}}) + \lambda_a \, \mathrm{KSD}(p(\boldsymbol{Z}), q(\widehat{\boldsymbol{Z}})). \tag{12}$$

Besides, the score model in Eq. (11) communicates among different clients to obtain the global distribution, making it possible to be reliable guidance of invariance among clients. This makes SAG a potential generalization approach for modeling feature invariance in the overall FL scenario, even acting warm-start for unseen clients. Therefore, FOOGD is capable of both local IN-C data generalization and consistent performance generalization of clients. The algorithm of SAG can be found in Appendix A Algo. 3.

## 4 Theoretical Discussion

In this section, we provide the error bound of modeling score model via $\mathrm{SM}^3\mathrm{D}$ in federated scenarios, and provide the error bound in Theorem 4.1. Besides, the federated training procedure of score model is the same with the main task model. This indicates that our federated learning convergence bound is unchanged, following [40]. We provide more theoretical details in Appendix B.

**Theorem 4.1** (Error Bound of Decentralized Score Matching via $\mathrm{SM}^3\mathrm{D}$). *Assume the original* $\mathrm{MMD}(\boldsymbol{Z}, \boldsymbol{Z}_{gen}) \leq C$ *for randomly initialized score model* $s_{\boldsymbol{\theta}}(\boldsymbol{z})$ *in Eq. (7), the score model achieves optimum and MMD decreases. By Lemma B.1, we can obtain the final error bound of global* $s_{\boldsymbol{\theta}}(\cdot)$ *as:*

$$\|s_{\boldsymbol{\theta}}(\boldsymbol{z}) - \nabla_{\boldsymbol{z}} \log p_{\mathcal{D}}(\boldsymbol{z})\|^2 \leq \frac{\boldsymbol{v}^\top \boldsymbol{v}}{\sigma^2} - \mathbb{E}_{p_{\mathcal{D}}(\boldsymbol{z})}[\|\nabla_{\boldsymbol{z}} \log p_{\mathcal{D}}(\boldsymbol{z})\|^2] + \frac{|\mathcal{D}|}{B} C, \tag{13}$$

*where* $C$ *is the upper bound of the MMD,* $B$ *is batch size, and* $|\mathcal{D}|$ *is the data amount.*

## 5 Experiments

### 5.1 Experimental Setups

**Datasets.** Following SCONE [3], we choose clear Cifar10, Cifar100 [33], and TinyImageNet [36] as the IN data, and select the corresponding corrupted versions [19], i.e., Cifar10-C, Cifar100-C and TinyImageNet-C as IN-C data. We evaluate detection with five OUT image datasets: SVHN [59], Texture [11], iSUN [78], LSUN-C and LSUN-R [81]. To simulate the non-IID scenarios, we sample data by label in a Dirichlet distribution parameterized by non-IID degree [23], i.e., $\alpha$, for $K$ clients. The smaller $\alpha$ simulates the more heterogeneous client data distribution in federated settings. To evaluate FOOGD on unseen client generalization data, we also use PACS [38] dataset for leave-one-out domain generalization. Details of dataset simulation are in Appendix D.1.

**Comparison Methods and Evaluations.** We study the performance of FOOGD with the state-of-the-art (SOTA) federated learning model and FedAvg-like derivant of SOTA centralized OOD methods, i.e., LogitNorm [76] (FedLN), ATOL [88] (FedATOL), T3A [25] (FedT3A). We compare FOOGD with three types of baseline models, i.e., (1) *Vanilla FL model*: **FedAvg** [56] and **FedRoD** [8], (2) *FL with OOD detection*: **FOSTER** [83], **FedLN**, and **FedATOL**, (3) *FL with OOD generalization*: **FedT3A**, **FedIIR** [17], **FedTHE** [26], **FedICON** [69]. For evaluation, we report the accuracy of IN data (ACC-IN) and IN-C data (ACC-IN-C) to validate IN generalization and OOD generalization, respectively. We compute the maximum softmax probability [20] (MSP) and report the standard metrics used for OOD detection, i.e., the area under the receiver operating characteristic curve (AUROC), and the false positive rate at threshold corresponding to a true positive rate of 95% (FPR95) [20].

**Implementation Details.** We choose WideResNet [85] as our main task model for Cifar datasets, and ResNet18 [18] for TinyImageNet and PACS, and optimize each model 5 local epochs per communication round until converging with SGD optimizer. We conduct all methods at their best

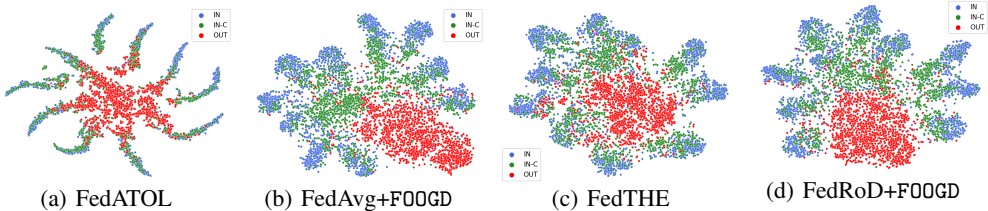

| (a) FedATOL | (b) FedAvg+FOOGD | (c) FedTHE | (d) FedRoD+FOOGD |

Figure 4: T-SNE visualizations of FedAvg and FedRoD with FOOGD.

and report the average results of three repetitions with different random seeds. We consider client number $K = 10$, participating ratio of $1.0$ for performance comparison, and the hyperparameters $\lambda_m = 0.5$, $\lambda_a = 0.05$. We provide the full implementation details in Appendix D.2.

Table 1: Main results of federated OOD detection and generalization on Cifar10. We report the ACC of brightness as IN-C ACC, the FPR95 and AUROC of LSUN-C as OUT performance.

| Non-IID | $\alpha = 0.1$ | | | | $\alpha = 0.5$ | | | | $\alpha = 5.0$ | | | |
| Method | ACC-IN↑ | ACC-IN-C↑ | FPR95↓ | AUROC↑ | ACC-IN↑ | ACC-IN-C↑ | FPR95↓ | AUROC↑ | ACC-IN↑ | ACC-IN-C↑ | FPR95↓ | AUROC↑ |
|---|---|---|---|---|---|---|---|---|---|---|---|---|
| FedAvg | 68.03 | 65.44 | 83.41 | 58.05 | 86.59 | 83.72 | 43.70 | 84.18 | 86.50 | 85.08 | 38.24 | 85.37 |
| FedLN | 75.24 | 71.77 | 56.14 | 84.14 | 86.10 | 84.20 | 39.26 | 89.64 | 87.20 | 85.08 | 33.33 | 90.87 |
| FedATOL | 55.93 | 54.44 | 49.50 | 86.22 | 87.55 | 85.64 | 27.87 | 93.48 | 89.27 | 88.28 | 19.66 | 95.25 |
| FedT3A | 68.03 | 61.52 | 78.12 | 63.64 | 86.59 | 82.85 | 43.70 | 84.18 | 86.50 | 85.01 | 38.24 | 85.37 |
| FedIIR | 68.26 | 66.12 | 79.48 | 63.31 | 86.75 | 84.75 | 40.91 | 84.94 | 87.77 | 86.10 | 34.69 | 87.66 |
| FedAvg+FOOGD | 75.09 | 73.71 | 35.32 | 91.21 | 88.36 | 87.26 | 17.78 | 96.53 | 88.90 | 88.25 | 12.02 | 97.77 |
| FedRoD | 91.15 | 89.90 | 47.97 | 80.96 | 89.62 | 87.70 | 37.03 | 86.50 | 87.69 | 86.26 | 36.13 | 86.65 |
| FOSTER | 90.22 | 88.70 | 47.40 | 77.43 | 86.92 | 85.82 | 42.03 | 83.91 | 87.83 | 85.96 | 36.42 | 86.19 |
| FedTHE | 91.05 | 89.71 | 58.14 | 82.04 | 89.14 | 87.68 | 40.28 | 85.30 | 88.14 | 86.18 | 35.35 | 86.79 |
| FedICON | 89.06 | 89.18 | 48.22 | 81.28 | 75.83 | 75.35 | 56.19 | 79.88 | 87.20 | 85.39 | 35.63 | 86.45 |
| FedRoD+FOOGD | 93.51 | 92.74 | 32.99 | 91.76 | 90.46 | 90.16 | 25.51 | 94.19 | 89.44 | 88.62 | 18.91 | 96.25 |

## 5.2 Experimental Results

**Performance Comparison on non-IID data.** We categorized our baseline models into two groups based on whether they consider personalization. The results for Cifar10, Cifar100, and TinyImageNet are shown in Tab. 1, Tab. 2, and Tab. 7 in Appendix E.1, respectively. **For the first group without considering personalization**, the existing centralized OOD methods, i.e., LogitNorm (FedLN), ATOL (FedATOL) and T3A (FedT3A), are not directly competitive among different non-IID scenarios. Though FedATOL achieves satisfying results for both generalization and detection tasks on Cifar10 $\alpha = 5$, it fails neither in smaller $\alpha$ and dataset containing more classes (i.e., Cifar100). Meanwhile, the vanilla FedAvg degrades its performance in OOD generalization for both Cifar10 and Cifar100 data, and shows no potential of detecting OUT data samples. FedIIR pays more effort to maintain the inter-client generalization via restricting model consistency, making it less effective in non-IID settings. **For the second group of personalized FL methods**, personalization is quite necessary for both IN data generalization and IN-C generalization, which is similarly illustrated in FedTHE [26] and FedICON [69]. In general, personalized methods are worse in FL detection than non-personalized methods, indicating that there is a conflict between detecting OUT data and enhancing prediction in non-IID setting. More surprisingly, we also discover that personalized adaption methods also detect outliers better compared with vanilla FedRoD model. FOSTER has better detection in more heterogenous data distribution, i.e., $\alpha = 0.1$, compared with its results in $\alpha = 5$, but its overall performance is supposed to enhance in the future. FOOGD **is a flexible FL**

Table 2: Main results of federated OOD detection and generalization on Cifar100. We report the ACC of brightness as IN-C ACC, the FPR95 and AUROC of LSUN-C as OUT performance.

| Non-IID | $\alpha = 0.1$ | | | | $\alpha = 0.5$ | | | | $\alpha = 5.0$ | | | |
| Method | ACC-IN↑ | ACC-IN-C↑ | FPR95↓ | AUROC↑ | ACC-IN↑ | ACC-IN-C↑ | FPR95↓ | AUROC↑ | ACC-IN↑ | ACC-IN-C↑ | FPR95↓ | AUROC↑ |
|---|---|---|---|---|---|---|---|---|---|---|---|---|
| FedAvg | 51.67 | 47.54 | 78.35 | 67.16 | 58.28 | 54.62 | 72.84 | 70.86 | 61.40 | 56.72 | 72.68 | 70.59 |
| FedLN | 52.48 | 48.15 | 66.94 | 74.82 | 59.39 | 53.86 | 68.31 | 73.41 | 61.00 | 56.33 | 69.18 | 75.87 |
| FedATOL | 43.65 | 41.08 | 65.26 | 81.64 | 60.62 | 56.63 | 70.10 | 79.27 | 64.16 | 63.61 | 80.27 | 60.51 |
| FedT3A | 51.67 | 51.50 | 78.36 | 67.22 | 59.07 | 55.42 | 72.86 | 70.88 | 61.64 | 55.51 | 72.77 | 70.44 |
| FedIIR | 51.63 | 47.88 | 81.91 | 63.99 | 58.66 | 55.72 | 77.62 | 65.87 | 61.70 | 57.65 | 72.57 | 69.07 |
| FedAvg+FOOGD | 53.84 | 51.69 | 36.40 | 91.41 | 61.82 | 59.91 | 55.70 | 86.42 | 64.96 | 64.18 | 57.70 | 84.03 |
| FedRoD | 73.13 | 69.26 | 66.34 | 73.02 | 66.88 | 61.28 | 70.13 | 69.48 | 61.34 | 55.80 | 74.86 | 67.76 |
| FOSTER | 72.54 | 67.50 | 61.25 | 75.44 | 62.45 | 57.62 | 73.26 | 68.71 | 53.80 | 49.28 | 76.94 | 65.47 |
| FedTHE | 73.83 | 69.09 | 64.73 | 75.16 | 66.22 | 61.19 | 72.95 | 69.38 | 61.03 | 57.03 | 71.43 | 69.01 |
| FedICON | 72.22 | 67.79 | 61.36 | 77.12 | 65.86 | 61.83 | 69.99 | 71.03 | 62.11 | 57.62 | 70.91 | 70.84 |
| FedRoD+FOOGD | 77.88 | 75.70 | 58.81 | 86.07 | 70.30 | 68.23 | 45.19 | 89.59 | 64.94 | 62.56 | 65.18 | 80.47 |

Table 3: Cifar10 ablation study on varying $\alpha$ modeled by FedAvg.

| Non-IID | $\alpha = 0.1$ | | | | $\alpha = 0.5$ | | | | $\alpha = 5.0$ | | | |
|---|---|---|---|---|---|---|---|---|---|---|---|---|
| Method | ACC-IN↑ | ACC-IN-C↑ | FPR95↓ | AUROC↑ | ACC-IN↑ | ACC-IN-C↑ | FPR95↓ | AUROC↑ | ACC-IN↑ | ACC-IN-C↑ | FPR95↓ | AUROC↑ |
| fix backbone | 68.03 | 65.44 | 51.27 | 88.49 | 86.59 | 83.72 | 20.40 | 95.82 | 86.50 | 85.08 | 15.44 | 96.96 |
| w/o SM$^3$D | 74.70 | 73.35 | 41.86 | 88.88 | 88.01 | 87.17 | 19.96 | 95.86 | 88.52 | 87.79 | 15.05 | 97.06 |
| w/o SAG | 73.15 | 70.79 | 37.59 | 91.47 | 87.32 | 85.33 | 18.83 | 96.13 | 87.86 | 86.20 | 12.73 | 97.65 |
| FedAvg+FOOGD | **75.09** | **73.71** | **35.32** | **91.21** | **88.36** | **87.26** | **17.78** | **96.53** | **88.90** | **88.25** | **12.02** | **97.77** |

Table 4: Cifar100 ablation study on varying $\alpha$ modeled by FedAvg.

| Non-IID | $\alpha = 0.1$ | | | | $\alpha = 0.5$ | | | | $\alpha = 5.0$ | | | |
|---|---|---|---|---|---|---|---|---|---|---|---|---|
| Method | ACC-IN↑ | ACC-IN-C↑ | FPR95↓ | AUROC↑ | ACC-IN↑ | ACC-IN-C↑ | FPR95↓ | AUROC↑ | ACC-IN↑ | ACC-IN-C↑ | FPR95↓ | AUROC↑ |
| fix backbone | 51.67 | 47.54 | 56.11 | 82.94 | 58.28 | 54.62 | 68.90 | 77.26 | 61.40 | 56.72 | 68.04 | 77.05 |
| w/o SM$^3$D | 53.45 | 51.58 | 43.49 | 89.26 | 61.82 | 59.91 | 62.18 | 84.72 | 64.03 | 62.19 | 64.18 | 83.16 |
| w/o SAG | 53.14 | 48.35 | 37.23 | 91.13 | 60.39 | 55.72 | 60.53 | 85.17 | 62.12 | 57.16 | 59.58 | 82.84 |
| FedAvg+FOOGD | **53.84** | **51.69** | **36.40** | **91.41** | **62.19** | **60.25** | **55.70** | **86.42** | **64.96** | **64.18** | **57.70** | **84.03** |

**framework and achieves significant results for wild data tasks, i.e., IN generalization, IN-C generalization, and OUT detection, on Cifar10, Cifar100, and TinyImageNet.** Specifically, FOOGD achieves comparable performance in enhancing both FedAvg and FedRoD, free of the FL framework constraints. FOOGD enjoys the benefits of SM$^3$D, achieving distinguishable detection improvement by Eq. (9). Besides, the regularization of score model with global distribution makes SAG regularize main task feature extractor better than contrastive-based methods, e.g., FedICON, and rebalanced-methods, e.g., FedTHE and original FedRoD.

**Ablation Studies.** We devise the variants of FOOGD, i.e., fix backbone, w/o SM$^3$D, and w/o SAG, to study the effectiveness of our three main ideas: (1) obtaining reliable global distribution as guidance, (2) estimating score model by SM$^3$D, and (3) enhancing FL method generalization by SAG, respectively. From Tab. 3 and Tab. 4, simply modeling score model enhances detection slightly, since it brings the knowledge of global distribution. When we remove SM$^3$D, the estimation of data probability is severely impacted, bringing no detection capability. While the generalization performance decreases once we remove SAG. Moreover, compared with fix backbone, both w/o SM$^3$D and w/o SAG have better generalization and detection results, indicating the necessity of regularizing feature extractor with global distribution.

**Visualization.** To explore the wild data distribution of FL OOD methods, we visualize T-SNE of data representations in Fig. 4, and the detection score distributions in Fig. 5, on Cifar10 $\alpha = 5$ for FedAvg+FOOGD, FedRoD+FOOGD and their runner-up methods, FedATOL and FedTHE, respectively. It is evident that FOOGD represents IN-C data more tight with IN data, and constructs a comparably clear decision boundary between IN data and OUT data. Besides, we also discover that FOOGD will push OUT data away from its IN and IN-C data, which validates the guidance from the global distribution. Additionally, in Fig. 5, FOOGD makes the modes among IN, IN-C, and OUT, more separable than existing methods. This also proves the effectiveness of FOOGD in detection task.

**Extensive experiments on other IN-C and OUT data.** In this part, we study the performance evaluation of FOOGD in additional IN-C and OUT datasets. In Tab. 5, we can find that FOOGD consistently enhances the detection capability for different OUT data, validating the effectiveness of estimating global distribution via SM$^3$D. Meanwhile, we compute the average results of different IN-C accuracy for FL models trained on Cifar10 and Cifar100 in Fig. 6. *The +FOOGD in each group is short for FedAvg+FOOGD and FedRoD+FOOGD, respectively.* We provide the details in Appendix E.7 Tab. 13 and Tab 14. FOOGD consistently improve the generalization in all unseen IN-C data, indicating the effectiveness of enhancing feature extractor via SAG.

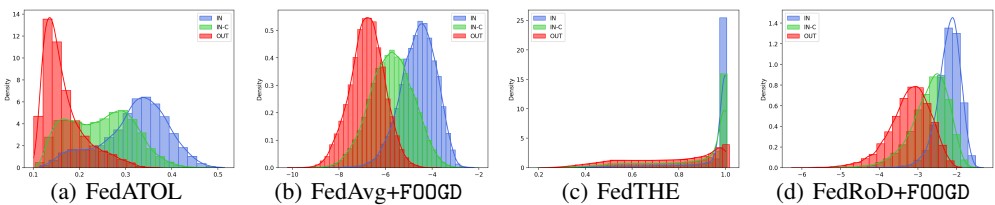

| (a) FedATOL | (b) FedAvg+FOOGD | (c) FedTHE | (d) FedRoD+FOOGD |

Figure 5: Detection score distribution of FL methods on Cifar10 ($\alpha = 5.0$).

Table 5: Other detection results on Cifar10 ($\alpha = 0.1$).

| OUT Data | iSUN | | SVHN | | LSUN-R | | Texture | |
|---|---|---|---|---|---|---|---|---|
| Method | FPR95↓ | AUROC↑ | FPR95↓ | AUROC↑ | FPR95↓ | AUROC↑ | FPR95↓ | AUROC↑ |
| FedAvg | 62.10 | 76.29 | 80.02 | 62.14 | 62.01 | 77.02 | 80.53 | 66.23 |
| FedLN | 66.41 | 76.03 | 70.95 | 76.82 | 61.31 | 78.34 | 93.90 | 71.99 |
| FedATOL | 61.01 | 80.05 | 85.39 | 82.17 | 64.01 | 79.89 | 66.33 | 78.77 |
| FedIIR | 57.86 | 77.98 | 83.68 | 64.04 | 58.44 | 78.69 | 91.72 | 62.32 |
| FedAvg+FOOGD | 37.55 | 91.22 | 44.59 | 87.63 | 44.16 | 90.16 | 28.60 | 91.75 |
| FedRoD | 43.40 | 82.83 | 40.72 | 83.55 | 41.80 | 82.92 | 53.24 | 81.52 |
| FOSTER | 48.73 | 76.29 | 39.55 | 83.07 | 48.09 | 76.24 | 54.23 | 77.62 |
| FedTHE | 43.72 | 83.50 | 39.22 | 85.95 | 42.95 | 83.46 | 53.58 | 82.19 |
| FedICON | 49.98 | 82.95 | 34.94 | 85.56 | 49.05 | 83.30 | 51.57 | 80.96 |
| FedRoD+FOOGD | 36.17 | 88.69 | 17.61 | 94.56 | 41.46 | 92.80 | 19.46 | 93.39 |

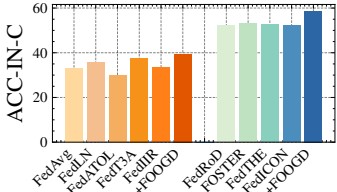

Figure 6: The average results for Cifar100-C generalization.

**Client Generalization on PACS Dataset.** To validate the effectiveness of FOOGD in domain generalization tasks, i.e., each client contains one domain data and we train domain generalization model by leave-one-out, following FedIIR [17]. To compare fairly, we pretrain all models from scratch and utilize adaption methods as stated in their main paper. In terms of Tab. 6, FOOGD obtains performance improvements for FedAvg and FedRoD. Compared with existing adaption methods, FOOGD achieves outstanding results even in the toughest task, i.e., leaving Sketch domain out. This also concludes that FOOGD is capable of inter-client generalization, via utilizing global distribution knowledge.

**Hyperparameter sensitivity studies and other empirical studies.** Due to the space limitation, we leave the other relevant experiments in Appendix E. Summarily, we study four additional evaluations: (1) In Tab. 10 We compute different detection metrics, i.e., MSP, energy score, and ASH, and validate that Eq. (9) is consistently powerful in detection. (2) We

Table 6: OOD generalization task for PACS.

| Method \Domain | Art Painting | Cartoon | Photo | Sketch | Average |
|---|---|---|---|---|---|
| FedAvg | 97.21 | 62.58 | 91.00 | 35.28 | 71.52 |
| FedRoD | 93.45 | 88.85 | 89.34 | 29.95 | 75.39 |
| FedT3A | 97.13 | 75.71 | 93.21 | 37.40 | 75.86 |
| FedIIR | 86.86 | 80.29 | 88.98 | 31.38 | 71.88 |
| FedTHE | 96.17 | 90.72 | 93.57 | 29.14 | 77.40 |
| FedICON | 50.42 | 53.36 | 52.19 | 50.87 | 51.58 |
| FedAvg+FOOGD | 97.46 | 89.32 | 91.48 | 41.40 | 79.92 |
| FedRoD+FOOGD | 97.85 | 92.31 | 93.01 | 50.95 | 83.53 |

vary the coefficient of SM³D $\lambda_m = \{0.1, 0.2, 0.5, 0.8, 1\}$ in Fig. 11(a)-Fig. 11(b), and vary the coefficient of SAG $\lambda_a = \{0, 0.01, 0.05, 0.1, 0.2, 0.5, 0.8\}$ in Fig. 11(c), to obtain the best modeling in FOOGD. (3) We vary the number of participating clients in Fig. 10 and found FOOGD can have better results among different participating clients.

## 6 Conclusion and Future Work

In this work, we consider enhancing both detection and generalization capability of FL methods among non-IID settings. To realize it, we try to model global distribution by collaborating clients, and propose FOOGD, which consists of SM³D for estimating score model for detection, and SAG to enhance the invariant representation for generalization. We conduct extensive experiments to validate the effectiveness of FOOGD: (1) reliably and flexibly estimating non-normalized decentralized distribution, (2) detecting semantic shift data via the norm of score values, and (3) generalizing adaption of covariate shift data by regularizing feature extractor invariant distribution discrepancy.

In the future, we plan to integrate privacy enhancement techniques, such as differential privacy, into FOOGD. While the score model in FOOGD captures the score function of the data probability in the latent space, which is extremely difficult to be used for reconstructing the original data by attacking. The primary risk exposure for each client arises from the exchange of model parameters, i.e., the feature extractor and score model. Hence, FOOGD has a comparable level of privacy exposure as existing FL methods dealing with non-IID and OOD shifts, acquiring to be addressed comprehensively.

## Acknowledgments and Disclosure of Funding

This work was supported in part by the Leading Expert of "Ten Thousands Talent Program" of Zhejiang Province, China (No. 2021R52001), the National Natural Science Foundation of China (No. 62172362), and the Fundamental Research Funds for the Central Universities (No. 226202400241).

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

## A    Algorithms

The overall algorithm of `FOOGD` is in Algo. 1. In line 1:10, the server collaborates with clients to optimize the feature extractor model for representation and the score model for density estimation. The clients execute training local models separately in line 11:19. In each client, `SM`$^3$`D` estimates data density based on the latent representation of feature extractor, and `SAG` computes the kernelized stein discrepancy based on score model to regularize the optimization of feature extractor. We update score model with `SM`$^3$`D` in Algo. 2, and the training procedure of feature extractor is detailed in Algo. 3.

---

**Algorithm 1** Training procedure of `FOOGD`

---

**Input**: Batch size $B$, communication rounds $T$, number of clients $K$, local steps $E$, dataset $\mathcal{D} = \cup_{k \in [K]} \mathcal{D}_k$

**Output**: feature extractor and score model parameters, i.e., $\boldsymbol{\theta}_f^T$ and $\boldsymbol{\theta}_s^T$

1: **Server executes():**
2: Initialize $\{\boldsymbol{\theta}_f^0, \boldsymbol{\theta}_s^0\}$ with random distribution
3: **for** $t = 0, 1, ..., T - 1$ **do**
4:     **for** $k = 1, 2, ..., K$ **in parallel do**
5:         Send $\{\boldsymbol{\theta}_f^t, \boldsymbol{\theta}_s^t\}$ to client $k$
6:         $\{\boldsymbol{\theta}_f^{t,k}, \boldsymbol{\theta}_s^{t,k}\} \leftarrow$ **Client executes**$(k, \{\boldsymbol{\theta}_f^t, \boldsymbol{\theta}_s^t\})$
7:     **end for**
8:     Update parameters of $\{\boldsymbol{\theta}_f^t, \boldsymbol{\theta}_s^t\}$ by Eq. (3)
9: **end for**
10: **return** $\{\boldsymbol{\theta}_f^T, \boldsymbol{\theta}_s^T\}$
11: **Client executes**$(k, \{\boldsymbol{\theta}_f^t, \boldsymbol{\theta}_s^t\})$:
12: Assign global model to local model $\{\boldsymbol{\theta}_f^k, \boldsymbol{\theta}_s^k\} \leftarrow \{\boldsymbol{\theta}_f^t, \boldsymbol{\theta}_s^t\}$
13: **for** each local epoch $e = 1, 2, ..., E$ **do**
14:     **for** batch of samples $(\boldsymbol{X}_{1:B}, \boldsymbol{Y}_{1:B}) \in \mathcal{D}_k$ **do**
15:         Execute $\boldsymbol{\theta}_s^k = $ `SM`$^3$`D`$(\boldsymbol{X}_{1:B}, \boldsymbol{Y}_{1:B})$ in Algorithm 2
16:         Execute $\boldsymbol{\theta}_f^k = $ `SAG`$(\boldsymbol{X}_{1:B}, \boldsymbol{Y}_{1:B})$ in Algorithm 3
17:     **end for**
18: **end for**
19: **return** $\boldsymbol{\theta}_k^E$ to server

---

**Algorithm 2** Algorithm of `SM`$^3$`D`

---

**Input**: Batch size $B$, batch of samples $(\boldsymbol{X}_{1:B}, \boldsymbol{Y}_{1:B}) \in \mathcal{D}_k$, fixed feature extractor $\boldsymbol{\theta}_f$, and initialized score model $\boldsymbol{\theta}_s$

**Output**: score model parameters, i.e., $\boldsymbol{\theta}_s$

1: Feature Extraction $\boldsymbol{Z}_{1:B} \leftarrow f_{\boldsymbol{\theta}}(\boldsymbol{X}_{1:B})$
2: Sample $B$ random data points $\boldsymbol{Z}^0$ with $z_i^0 \sim \mathcal{N}(\boldsymbol{0}, \boldsymbol{I})$
3: Take Langevin dynamic sampling started at $\boldsymbol{Z}^0$ to obtain generated samples $\boldsymbol{Z}_{\text{gen}}$ by Eq. (6)
4: Perturb noise on data features to obtain $\tilde{\boldsymbol{Z}} \sim \mathcal{N}(\boldsymbol{Z}, \sigma \boldsymbol{I})$
5: Compute denoising score matching by Eq. (4)
6: Regularize maximum mean discrepancy between generated $\boldsymbol{Z}_{\text{gen}}$ and $\boldsymbol{Z}$ by Eq. (7)
7: Optimize score model $\boldsymbol{\theta}_s$ with objective $\ell_k^{\text{OUT}}$ by Eq. (8)
8: **return** $\boldsymbol{\theta}_s$

---

**Algorithm 3** Algorithm of SAG

**Input**: Batch size $B$, batch of samples $(\boldsymbol{X}_{1:B}, \boldsymbol{Y}_{1:B}) \in \mathcal{D}_k$, fixed score model $\boldsymbol{\theta}_s$, and initialized feature extractor $\boldsymbol{\theta}_f$
**Output**: feature extractor parameters, i.e., $\boldsymbol{\theta}_f$

1: Augment samples $\widehat{\boldsymbol{X}}_{1:B} = \mathcal{T}(\boldsymbol{X}_{1:B})$
2: Feature extraction $\boldsymbol{Z}_{1:B} \leftarrow f_{\boldsymbol{\theta}}(\boldsymbol{X}_{1:B})$, and $\widehat{\boldsymbol{Z}}_{1:B} \leftarrow f_{\boldsymbol{\theta}}(\widehat{\boldsymbol{X}}_{1:B})$
3: Compute kernelized Stein divergence by Eq. (11)
4: Compute cross entropy loss between prediction $\boldsymbol{Y}_{pred} = \text{Classifier}(\boldsymbol{Z})$ and ground truth $\boldsymbol{Y}_{gr}$
5: Optimize feature extractor $\boldsymbol{\theta}_f$ with objective $\ell_k^{\text{IN}} + \ell_k^{\text{IN-C}}$ by Eq. (12)
6: **return** $\boldsymbol{\theta}_f$

# B Theoretical Analysis

## B.1 Error Bound of SM³D

**Lemma B.1** (Error Bound of Decentralized Score Matching). *The error bound of global score model aggregated from local scores models is*

$$\|\nabla_{\boldsymbol{z}} \log p_{\boldsymbol{\theta}}(\boldsymbol{z}) - \nabla_{\boldsymbol{z}} \log p_{\mathcal{D}}(\boldsymbol{z})\|^2 = \frac{\boldsymbol{v}^{\top}\boldsymbol{v}}{\sigma^2} - \mathbb{E}_{p_{\mathcal{D}}(\boldsymbol{z})}[\|\nabla_{\boldsymbol{z}} \log p_{\mathcal{D}}(\boldsymbol{z})\|^2]. \tag{14}$$

*Proof.* For the global score model aggregated from local score models that estimate IN data probability densities, it holds:

$$\begin{aligned}
\nabla_{\boldsymbol{z}} \log p_{\boldsymbol{\theta}}(\boldsymbol{z}) = s_{\boldsymbol{\theta}}(\boldsymbol{z}) &= \sum_{k=1}^{K} w_k s_{\boldsymbol{\theta}_k}(\boldsymbol{z}) \\
&= \sum_{k=1}^{K} w_k \nabla_{\boldsymbol{z}} \log p_{\boldsymbol{\theta}_k}(\boldsymbol{z}) \\
&= \nabla_{\boldsymbol{z}} \sum_{k=1}^{K} w_k \log p_{\boldsymbol{\theta}_k}(\boldsymbol{z}).
\end{aligned} \tag{15}$$

Then we formulate the score matching for global distribution as

$$\begin{aligned}
&\|\nabla_{\boldsymbol{z}} \log p_{\boldsymbol{\theta}}(\boldsymbol{z}) - \nabla_{\boldsymbol{z}} \log p_{\mathcal{D}}(\boldsymbol{z})\|^2 \\
&= \|\sum_{k=1}^{K} w_k \nabla_{\boldsymbol{z}} \log p_{\boldsymbol{\theta}_k}(\boldsymbol{z}) - \nabla_{\boldsymbol{z}} \log p_{\mathcal{D}}(\boldsymbol{z})\|^2 \\
&= \|\sum_{k=1}^{K} w_k \nabla_{\boldsymbol{z}} \log p_{\boldsymbol{\theta}_k}(\boldsymbol{z}) - \sum_{k=1}^{K} w_k \nabla_{\boldsymbol{z}} \log p_{\mathcal{D}}(\boldsymbol{z})\|^2 \\
&= \|\sum_{k=1}^{K} w_k [\nabla_{\boldsymbol{z}} \log p_{\boldsymbol{\theta}_k}(\boldsymbol{z}) - \nabla_{\boldsymbol{z}} \log p_{\mathcal{D}}(\boldsymbol{z})]\|^2 \\
&\leq \sum_{k=1}^{K} w_k \|\nabla_{\boldsymbol{z}} \log p_{\boldsymbol{\theta}_k}(\boldsymbol{z}) - \nabla_{\boldsymbol{z}} \log p_{\mathcal{D}}(\boldsymbol{z})\|^2,
\end{aligned} \tag{16}$$

where $\sum_{k=1}^{K} w_k = 1$, and the last term is held by Jensen inequation.

In term of Vincent [72], the DSM for each local model $s_{\theta_k}(z)$ is bounded as follows,

$$
\begin{aligned}
J_{\mathrm{DSM}}(\boldsymbol{\theta}_k) &\stackrel{\mathrm{def}}{=} \mathbb{E}_{p(\boldsymbol{z},\tilde{\boldsymbol{z}})}\left[\|s_{\boldsymbol{\theta}_k}(\boldsymbol{z}) - \nabla_{\tilde{\boldsymbol{z}}}\log p\left(\tilde{\boldsymbol{z}}\mid\boldsymbol{z}\right)\|^2\right] \\
&= \mathbb{E}_{p(\boldsymbol{z},\tilde{\boldsymbol{z}})}\left[\|s_{\boldsymbol{\theta}_k}(\boldsymbol{z})\|^2 - 2s_{\boldsymbol{\theta}_k}(\boldsymbol{z})^{\top}\nabla_{\tilde{\boldsymbol{z}}}\log p\left(\tilde{\boldsymbol{z}}\mid\boldsymbol{z}\right) + \|\nabla_{\tilde{\boldsymbol{z}}}\log p\left(\tilde{\boldsymbol{z}}\mid\boldsymbol{z}\right)\|^2\right] \\
&= \mathbb{E}_{p(\boldsymbol{z})}\left[\|s_{\boldsymbol{\theta}_k}(\boldsymbol{z})\|^2\right] - 2\mathbb{E}_{p(\boldsymbol{z},\tilde{\boldsymbol{z}})}\left[s_{\boldsymbol{\theta}_k}(\boldsymbol{z})^{\top}\nabla_{\tilde{\boldsymbol{z}}}\log p\left(\tilde{\boldsymbol{z}}\mid\boldsymbol{z}\right)\right] + \frac{\boldsymbol{v}^{\top}\boldsymbol{v}}{\sigma^2} \\
&= J_{\mathrm{ESM}}(\boldsymbol{\theta}_k) + 2\mathbb{E}_{p(\boldsymbol{z},\tilde{\boldsymbol{z}})}\left[s_{\boldsymbol{\theta}_k}(\boldsymbol{z})^{\top}\frac{\boldsymbol{v}}{\sigma}\right] + \frac{\boldsymbol{v}^{\top}\boldsymbol{v}}{\sigma^2} - \mathbb{E}_{p(\boldsymbol{z})}[\|\nabla_{\tilde{\boldsymbol{z}}}\log p(\boldsymbol{z})\|^2] \\
&= J_{\mathrm{ESM}}(\boldsymbol{\theta}_k) + \frac{\boldsymbol{v}^{\top}\boldsymbol{v}}{\sigma^2} - \mathbb{E}_{p(\boldsymbol{z})}[\|\nabla_{\tilde{\boldsymbol{z}}}\log p(\boldsymbol{z})\|^2],
\end{aligned}
\tag{17}
$$

where $\boldsymbol{v}\sim\mathcal{N}(\boldsymbol{0},\boldsymbol{I})$ and

$$
\nabla_{\tilde{\boldsymbol{z}}}\log p\left(\tilde{\boldsymbol{z}}\mid\boldsymbol{z}\right) = \nabla_{\tilde{\boldsymbol{z}}}\left[\log\frac{1}{\left(\sqrt{2\pi\sigma^2}\right)^d}\exp\left\{-\frac{\|\tilde{\boldsymbol{z}}-\boldsymbol{z}\|^2}{2\sigma^2}\right\}\right] = -\frac{\tilde{\boldsymbol{z}}-\boldsymbol{z}}{\sigma^2} = -\frac{\boldsymbol{v}}{\sigma}.
\tag{18}
$$

Therefore, when $\boldsymbol{\theta}=\boldsymbol{\theta}^*=\boldsymbol{\theta}_k^*$, we have $J_{\mathrm{ESM}}(\boldsymbol{\theta}) = J_{\mathrm{ESM}}(\boldsymbol{\theta}_k) = 0 \quad \forall k\in[K]$, the global score matching finally satisfies:

$$
\begin{aligned}
&\|s_{\boldsymbol{\theta}}(\boldsymbol{z}) - \nabla_{\boldsymbol{z}}\log p_{\mathcal{D}}(\boldsymbol{z})\|^2 \\
&= \|\nabla_{\boldsymbol{z}}\log p_{\boldsymbol{\theta}}(\boldsymbol{z}) - \nabla_{\boldsymbol{z}}\log p_{\mathcal{D}}(\boldsymbol{z})\|^2 \\
&\leq \sum_{k=1}^K w_k\|\nabla_{\boldsymbol{z}}\log p_{\boldsymbol{\theta}_k}(\boldsymbol{z}) - \nabla_{\boldsymbol{z}}\log p_{\mathcal{D}}(\boldsymbol{z})\|^2 \\
&= \frac{\boldsymbol{v}^{\top}\boldsymbol{v}}{\sigma^2} - \mathbb{E}_{p_{\mathcal{D}}(\boldsymbol{z})}[\|\nabla_{\boldsymbol{z}}\log p_{\mathcal{D}}(\boldsymbol{z})\|^2],
\end{aligned}
\tag{19}
$$

which holds due to $\sum_{k=1}^K w_k = 1$. $\qquad\square$

**Theorem B.2** (Error Bound of Decentralized Score Matching via SM³D). *Assume the original* $\mathrm{MMD}(\boldsymbol{Z},\boldsymbol{Z}_{gen})\leq C$ *for randomly initialized score model* $s_{\boldsymbol{\theta}}(\boldsymbol{z})$ *in Eq. (7), the score model achieves optimum and MMD decreases. By Lemma B.1, we can obtain the final error bound of global* $s_{\boldsymbol{\theta}}(\cdot)$ *as:*

$$
\|s_{\boldsymbol{\theta}}(\boldsymbol{z}) - \nabla_{\boldsymbol{z}}\log p_{\mathcal{D}}(\boldsymbol{z})\|^2 \leq \frac{\boldsymbol{v}^{\top}\boldsymbol{v}}{\sigma^2} - \mathbb{E}_{p_{\mathcal{D}}(\boldsymbol{z})}[\|\nabla_{\boldsymbol{z}}\log p_{\mathcal{D}}(\boldsymbol{z})\|^2] + \frac{|\mathcal{D}|}{B}C,
\tag{20}
$$

*where $C$ is the upper bound of the MMD, $B$ is batch size, and $|\mathcal{D}|$ is the data amount.*

## B.2 The overall induction of Kernelized Stein Discrepancy in `SAG`

Assume feature distributions $p(\boldsymbol{z})$ and $q(\hat{\boldsymbol{z}})$ are two bounded distributions satisfying $\lim_{\|\boldsymbol{z}\|\to\infty}p(\boldsymbol{z})\phi(\boldsymbol{z}) = 0$ and $\lim_{\|\hat{\boldsymbol{z}}\|\to\infty}q(\hat{\boldsymbol{z}})\phi(\hat{\boldsymbol{z}}) = 0$. And we denote the gradient of log density in $\boldsymbol{z}$ as $\nabla_{\hat{\boldsymbol{z}}}\log q(\hat{\boldsymbol{z}}) = \frac{\nabla_{\hat{\boldsymbol{z}}}q(\hat{\boldsymbol{z}})}{q(\hat{\boldsymbol{z}})}$.

**Lemma B.3** (Stein identity). *If the $\phi(\cdot)$ in Stein operator $\mathcal{A}_q\phi(\hat{\boldsymbol{z}}) = \phi(\hat{\boldsymbol{z}})\nabla_{\hat{\boldsymbol{z}}}\log q(\hat{\boldsymbol{z}}) + \nabla_{\hat{\boldsymbol{z}}}\phi(\hat{\boldsymbol{z}})$ introduced in Eq. (10) is Stein class, then we have a fundamental property called Stein identity as below:*

$$
\mathbb{E}_{\hat{\boldsymbol{z}}\sim q}\left[\phi(\hat{\boldsymbol{z}})\nabla_{\hat{\boldsymbol{z}}}\log q(\hat{\boldsymbol{z}}) + \nabla_{\hat{\boldsymbol{z}}}\phi(\hat{\boldsymbol{z}})\right] = 0.
\tag{21}
$$

*Proof.*

$$\mathbb{E}_{\widehat{z}\sim q}\left[\phi(\widehat{z})\nabla_{\widehat{z}}\log q(\widehat{z}) + \nabla_{\widehat{z}}\phi(\widehat{z})\right] = \int_{-\infty}^{+\infty} q(\widehat{z})\phi(\widehat{z})\nabla_{\widehat{z}}\log q(\widehat{z}) + q(\widehat{z})\nabla_{\widehat{z}}\phi(\widehat{z})\, d\widehat{z}$$

$$= \int_{-\infty}^{+\infty} q(\widehat{z})\phi(\widehat{z})\frac{\nabla_{\widehat{z}}q(\widehat{z})}{q(\widehat{z})} + q(\widehat{z})\nabla_{\widehat{z}}\phi(\widehat{z})\, d\widehat{z}$$

$$= \int_{-\infty}^{+\infty} \phi(\widehat{z})\nabla_{\widehat{z}}q(\widehat{z}) + q(\widehat{z})\nabla_{\widehat{z}}\phi(\widehat{z})\, d\widehat{z} \qquad (22)$$

$$= \int_{-\infty}^{+\infty} (\phi(\widehat{z})q(\widehat{z}))'d\widehat{z}$$

$$= \phi(\widehat{z})q(\widehat{z})|_{-\infty}^{+\infty}$$

$$= 0.$$

$\square$

**Definition B.4** (Stein Discrepancy). Stein identity induces Stein discrepancy for two distributions $p(z)$ and $q(\widehat{z})$:

$$\text{SD}(p(z), q(\widehat{z})) = \sup_{\phi\in\mathcal{F}} \mathbb{E}_{\widehat{z}\sim q}\left[\mathcal{A}_p\phi(\widehat{z})\right]^{\top} \mathbb{E}_{\widehat{z}'\sim q}\left[\mathcal{A}_p\phi(\widehat{z}')\right], \qquad (23)$$

where $\phi(\cdot)$ is the stein class function satisfying boundary conditions, and $\mathcal{F}$ is the function space.

**Lemma B.5.** *If $\mathcal{F}$ is a unit ball in reproducing kernel Hilbert space (RKHS) with positive definite kernel function $k(\cdot,\cdot) \in \mathcal{F}$, we obtain the Kernelized Stein Discrepancy for $p(z)$ and $q(\widehat{z})$ as below:*

$$\text{KSD}(p(z), q(\widehat{z})) = \mathbb{E}_{\widehat{z},\widehat{z}'\sim q}[s_{\boldsymbol{\theta}}(\widehat{z})^{\top}s_{\boldsymbol{\theta}}(\widehat{z}')k(\widehat{z},\widehat{z}') + s_{\boldsymbol{\theta}}(\widehat{z})^{\top}\nabla_{\widehat{z}'}k(\widehat{z},\widehat{z}') + s_{\boldsymbol{\theta}}(\widehat{z}')^{\top}\nabla_{\widehat{z}}k(\widehat{z},\widehat{z}')$$
$$+ \text{trace}(\nabla_{\widehat{z}}\nabla_{\widehat{z}'}k(\widehat{z},\widehat{z}'))]. \qquad (24)$$

*Proof.* Firstly, considering the expectation of $q(\widehat{z})$ on the Stein operator with score of $p(z)$, we can expand it via introducing Stein identity:

$$\mathbb{E}_{\widehat{z}\sim q}\left[\mathcal{A}_p\phi(\widehat{z})\right] = \mathbb{E}_{\widehat{z}\sim q}\left[\mathcal{A}_p\phi(\widehat{z})\right] - \mathbb{E}_{\widehat{z}\sim q}\left[\mathcal{A}_q\phi(\widehat{z})\right]$$
$$= \mathbb{E}_{\widehat{z}\sim q}\left[\mathcal{A}_p\phi(\widehat{z}) - \mathcal{A}_q\phi(\widehat{z})\right] \qquad (25)$$
$$= \mathbb{E}_{\widehat{z}\sim q}\left[\phi(\widehat{z})\left(\nabla_{\widehat{z}}\log p(\widehat{z}) - \nabla_{\widehat{z}}\log q(\widehat{z})\right)\right].$$

Then, with the property of RKHS, we have $k(\widehat{z},\widehat{z}') := \langle\phi(\widehat{z}),\phi(\widehat{z}')\rangle_{\mathcal{H}}$, $s_p(\widehat{z})$ and $s_q(\widehat{z})$ are short for $\nabla_{\widehat{z}}\log p(\widehat{z})$ and $\nabla_{\widehat{z}}\log q(\widehat{z})$, respectively, the Stein discrepancy can be rewritten as

$$\mathbb{S}(p(z), q(\widehat{z}))$$
$$= \mathbb{E}_{\widehat{z}\sim q}\left[\mathcal{A}_p\phi(\widehat{z})\right]^{\top}\mathbb{E}_{\widehat{z}'\sim q}\left[\mathcal{A}_p\phi(\widehat{z}')\right]$$
$$= \mathbb{E}_{\widehat{z},\widehat{z}'\sim q}\left[\left(\nabla_{\widehat{z}}\log p(\widehat{z}) - \nabla_{\widehat{z}}\log q(\widehat{z})\right)^{\top}k(\widehat{z},\widehat{z}')\left(\nabla_{\widehat{z}'}\log p(\widehat{z}') - \nabla_{\widehat{z}'}\log q(\widehat{z}')\right)\right]$$
$$= \mathbb{E}_{\widehat{z},\widehat{z}'\sim q}\left[\left(s_p(\widehat{z}) - s_q(\widehat{z})\right)^{\top}k(\widehat{z},\widehat{z}')\left(s_p(\widehat{z}') - s_q(\widehat{z}')\right)\right]$$
$$= \mathbb{E}_{\widehat{z},\widehat{z}'\sim q}\left[\left(s_p(\widehat{z}) - s_q(\widehat{z})\right)^{\top}\left(k(\widehat{z},\widehat{z}')s_p(\widehat{z}') + \nabla_{\widehat{z}'}k(\widehat{z},\widehat{z}') - k(\widehat{z},\widehat{z}')s_q(\widehat{z}') - \nabla_{\widehat{z}'}k(\widehat{z},\widehat{z}')\right)\right]$$
$$= \mathbb{E}_{\widehat{z},\widehat{z}'\sim q}\left[\left(s_p(\widehat{z}) - s_q(\widehat{z})\right)^{\top}\left(k(\widehat{z},\widehat{z}')s_p(\widehat{z}') + \nabla_{\widehat{z}'}k(\widehat{z},\widehat{z}')\right)\right], \qquad (26)$$

where the last second equation is also held by Stein identity.

Next, we define the

$$v(\widehat{z},\widehat{z}') = k(\widehat{z},\widehat{z}')s_p(\widehat{z}') + \nabla_{\widehat{z}'}k(\widehat{z},\widehat{z}'), \qquad (27)$$

introducing another Stein identity holds, i.e.,

$$\mathbb{E}_{\widehat{z},\widehat{z}'\sim q}\left[s_q(\widehat{z})^{\top}v(\widehat{z},\widehat{z}') + \nabla_{\widehat{z}}v(\widehat{z},\widehat{z}')\right] = \mathbb{E}_{\widehat{z},\widehat{z}'\sim q}\left[\mathcal{A}_q v(\widehat{z},\widehat{z}')\right] = 0. \qquad (28)$$

Finally, taking $v(\widehat{\boldsymbol{z}}, \widehat{\boldsymbol{z}}')$ back to Eq. (26) and substitute $s_{\boldsymbol{\theta}}(\widehat{\boldsymbol{z}}) = \nabla_{\widehat{\boldsymbol{z}}} \log p_{\boldsymbol{\theta}}(\widehat{\boldsymbol{z}})$, we can obtain the final KSD without the requirement of computing score values of $q(\widehat{\boldsymbol{z}})$.

$$
\begin{aligned}
&\text{KSD}(p(\boldsymbol{z}), q(\widehat{\boldsymbol{z}})) \\
&= \mathbb{E}_{\widehat{\boldsymbol{z}} \sim q} \left[ \mathcal{A}_p \phi(\widehat{\boldsymbol{z}}) \right]^\top \mathbb{E}_{\widehat{\boldsymbol{z}}' \sim q} \left[ \mathcal{A}_p \phi(\widehat{\boldsymbol{z}}') \right] \\
&= \mathbb{E}_{\widehat{\boldsymbol{z}}, \widehat{\boldsymbol{z}}' \sim q} \left[ \left( \nabla_{\widehat{\boldsymbol{z}}} \log p(\widehat{\boldsymbol{z}}) - \nabla_{\widehat{\boldsymbol{z}}} \log q(\widehat{\boldsymbol{z}}) \right)^\top k(\widehat{\boldsymbol{z}}, \widehat{\boldsymbol{z}}') \left( \nabla_{\widehat{\boldsymbol{z}}'} \log p(\widehat{\boldsymbol{z}}') - \nabla_{\widehat{\boldsymbol{z}}'} \log q(\widehat{\boldsymbol{z}}') \right) \right] \\
&= \mathbb{E}_{\widehat{\boldsymbol{z}}, \widehat{\boldsymbol{z}}' \sim q} \left[ s_{\boldsymbol{\theta}}(\widehat{\boldsymbol{z}})^\top s_{\boldsymbol{\theta}}(\widehat{\boldsymbol{z}}') k(\widehat{\boldsymbol{z}}, \widehat{\boldsymbol{z}}') + s_{\boldsymbol{\theta}}(\widehat{\boldsymbol{z}})^\top \nabla_{\widehat{\boldsymbol{z}}'} k(\widehat{\boldsymbol{z}}, \widehat{\boldsymbol{z}}') + s_{\boldsymbol{\theta}}(\widehat{\boldsymbol{z}}')^\top \nabla_{\widehat{\boldsymbol{z}}} k(\widehat{\boldsymbol{z}}, \widehat{\boldsymbol{z}}') + \text{trace}(\nabla_{\widehat{\boldsymbol{z}}} \nabla_{\widehat{\boldsymbol{z}}'} k(\widehat{\boldsymbol{z}}, \widehat{\boldsymbol{z}}')) \right].
\end{aligned}
\tag{29}
$$
$\square$

## B.3 Bound of Client Model Divergence

In this part, we first introduce mild and general assumptions [40], and induct the model updating divergence bound for each client. Because FOOGD aggregates client models similar to its original model, i.e., FedAvg and FedRoD, its generalization bound is unchanged compared with the generalization bound proposed in [40]. Please kindly refer to the original paper.

**Assumption B.6.** Let $F_k(\boldsymbol{\theta})$ be the expected model objective for client $k$, and assume $F_1, \cdots, F_K$ are all L-smooth, i.e., for all $\boldsymbol{\theta}_k$, $F_k(\boldsymbol{\theta}_k) \leq F_k(\boldsymbol{\theta}_k) + (\boldsymbol{\theta}_k - \boldsymbol{\theta}_k)^\top \nabla F_k(\boldsymbol{\theta}_k) + \frac{L}{2} \|\boldsymbol{\theta}_k - \boldsymbol{\theta}_k\|^2$.

**Assumption B.7.** Let $F_1, \cdots, F_N$ are all $\mu$-strongly convex: for all $\boldsymbol{\theta}_k$, $F_k(\boldsymbol{\theta}_k) \geq F_k(\boldsymbol{\theta}_k) + (\boldsymbol{\theta}_k - \boldsymbol{\theta}_k)^\top \nabla F_k(\boldsymbol{\theta}_k) + \frac{\mu}{2} \|\boldsymbol{\theta}_k - \boldsymbol{\theta}_k\|^2$.

**Assumption B.8.** Let $\xi_k^t$ be sampled from the $k$-th client's local data uniformly at random. The variance of stochastic gradients in each client is bounded: $\mathbb{E} \|\nabla F_k(\boldsymbol{\theta}_k^t, \xi_k^t) - \nabla F_k(\boldsymbol{\theta}_k^t)\|^2 \leq \sigma_k^2$.

**Assumption B.9.** The expected squared norm of stochastic gradients is uniformly bounded, i.e., $\mathbb{E} \|\nabla F_k(\boldsymbol{\theta}_k^t, \xi_k^t)\|^2 \leq V^2$ for all $k = 1, \cdots, K$ and $t = 1, \cdots, T-1$.

Next, we introduce the lemma related to the bound of client model divergence.

**Lemma B.10** (Bound of Client Model Divergence). *With assumption B.9, $\eta_t$ is non-increasing and $\eta_t < 2\eta_{t+E}$ (learning rate of t-th round and E-th epoch) for all $t \geq 0$, there exists $t_0 \leq t$, such that $t - t_0 \leq E - 1$ and $\boldsymbol{\theta}_k^{t_0} = \boldsymbol{\theta}^{t_0}$ for all $k \in [K]$. It follows that*

$$
\mathbb{E} \left[ \sum_{k=1}^K w_k \|\boldsymbol{\theta}^t - \boldsymbol{\theta}_k^t\|^2 \right] \leq 4\eta_t^2 (E-1)^2 V^2.
\tag{30}
$$

*Proof.* Let $E$ be the maximal local epoch. For any round $t > 0$, communication rounds from $t_0$ to $t$ exist $t - t_0 < E - 1$. and the global model $\boldsymbol{\theta}^{t_0}$ and each local model $\boldsymbol{\theta}_k^{t_0}$ are same at round $t_0$.

$$
\begin{aligned}
&\mathbb{E} \left[ \sum_{k=1}^K w_k \|\boldsymbol{\theta}^t - \boldsymbol{\theta}_k^t\|^2 \right] \\
&= \mathbb{E} \left[ \sum_{k=1}^K w_k \|(\boldsymbol{\theta}_k^t - \boldsymbol{\theta}^{t_0}) - (\boldsymbol{\theta}^t - \boldsymbol{\theta}^{t_0})\|^2 \right] &&(31a) \\
&\leq \mathbb{E} \left[ \sum_{k=1}^K w_k \|\boldsymbol{\theta}_k^t - \boldsymbol{\theta}^{t_0}\|^2 \right] &&(31b) \\
&= \mathbb{E} \left[ \sum_{k=1}^K w_k \left\| \sum_{\tau=t_0}^{t-1} \eta_t \nabla F_k(\boldsymbol{\theta}_k^\tau, \xi_k^\tau) \right\|^2 \right] &&(31c) \\
&\leq \mathbb{E} \left[ \sum_{k=1}^K w_k (t-t_0) \sum_{\tau=t_0}^{t-1} \eta_{t_0}^2 \|\nabla F_k(\boldsymbol{\theta}_k^\tau, \xi_k^\tau)\|^2 \right] &&(31d) \\
&\leq 4\eta_t^2 (E-1)^2 V^2, &&(31e)
\end{aligned}
$$

where the Eq. (31b) holds since $\mathbb{E}(\boldsymbol{\theta}_k^t - \boldsymbol{\theta}^{t_0}) = \boldsymbol{\theta}^t - \boldsymbol{\theta}^{t_0}$, and $\mathbb{E}\|X - \mathbb{E}(X)\| \leq \mathbb{E}\|X\|$, and Eq. (31d) derives from Jensen inequality. $\square$

## C  Related Work

### C.1  Federated Learning with Non-IID Data

Federated Learning (FL) with non-IID data presents significant challenges in balancing global and local model performance. One prominent method, FedAvg [56], uses simple averaging but struggles with client heterogeneity, often degrading individual client models. To improve global performance, methods like FedProx [39] introduce regularization to keep local updates close to the global model, and SCAFFOLD [30] uses control variates to reduce variance from client heterogeneity. For local personalization, meta-learning and transfer learning techniques such as DFL [52] focus on enhancing individual client models to adapt better to local data. Lastly, methods like FedRoD [8] attempt to achieve joint global and local performance by decomposing models, aiming to balance both objectives, though extreme non-IID settings still pose challenges. However, these FL methods modeling non-IID data take no actions to OOD data, causing them less advantageous.

## D  Experimental Implementation Details

### D.1  Experimental Setups

**Datasets**   Following SCONE [3], we choose clear TinyImageNet [36], Cifar10 and Cifar100 [33]and as the IN data. For OOD *generalization*, we select the corresponding synthetic covariate-shift dataset as IN-C data, by leveraging 15 common corruptions for all datasets, and 4 additional corruptions for Cifar10-C and Cifar100-C [19]. To evaluate `FOOGD` on unseen client data, we also perform experiments on PACS [38] for leave-one-out domain generalization. For OOD *detection*, we evaluate five OUT image datasets: SVHN [59], Texture [11], iSUN [78], LSUN-C and LSUN-R [81].

**Heterogeneous client data**   The original train and test datasets are split to all clients to simulate the practical non-IID scenario [23]. Specifically, we sample a proportion of instances of class $j$ to client $k$ using a Dirichlet distribution, i.e., $p_{j,k} \sim \text{Dir}(\alpha)$, where $\alpha$ denotes the non-IID degree of each class among the clients. A smaller $\alpha$ indicates a more heterogeneous data distribution. For the PACS dataset, 3 clients are set where each client holds data from one distinct domain, and the remaining unseen domain data is used for testing.

**OOD evaluation setups**   We construct three types of test sets to assess the model's classification, domain generalization, and out-of-distribution detection ability. Test set from IN dataset is used to evaluate how well the model adapts to local training distribution, i.e., model's classification ability. To simulate the non-IID distribution in real-world scenarios, we partition the IN-C dataset with the same heterogeneous distribution as the IN dataset. This setup evaluates the model's generalization ability on the IN-C dataset to determine whether existing FL methods can keep the data-label relationship in the presence of covariate shift in data features. All covariate-shift types in the IN-C dataset are test individually. For testing OOD detection ability, all clients use the same OOD test set for fair evaluation. After collecting performance data from all clients, we calculate a weighted average of the performance based on the volume of data each client holds.

### D.2  Implemetnation Details

We choose WideResNet [85] as our main task model for Cifar datasets, and ResNet18 [18] for TinyImageNet and PACS, and optimize each model 5 local epochs per communication round until converging with SGD optimizer. We conduct all methods at their best and report the average results of three repetitions with different random seeds. We consider client number $K = 10$, participating ratio of 1.0 for performance comparison, and the hyperparameters $\lambda_m = 0.5$, $\lambda_a = 0.05$.

Below are the detailed settings and hyperparameters for all federated baseline models.

1. **FedAvg** [56] is the classic federated learning method in which clients perform multiple epochs of SGD on their local data. The learning rate is set to 0.1, with a momentum of 0.9 and weight decay of 5e-4.

2. **FedIIR** [17] tries to implicitly learn invariant relationships through inter-client gradient alignment. We set the ema parameter 0.95 and penalty term 1e-3.

Table 7: Main results of federated OOD detection and generalization on TinyImageNet.

| Method | ACC-IN ↑ | ACC-IN-C↑ | FPR95↓ | AUROC↑ |
|---|---|---|---|---|
| FedAvg | 29.51 | 15.18 | 69.22 | 80.17 |
| FedLN | 38.23 | 15.64 | 61.40 | 82.31 |
| FedATOL | 23.32 | 13.69 | 29.04 | 94.91 |
| FedT3A | 29.46 | 00.50 | 69.14 | 80.08 |
| FedIIR | 38.01 | 14.90 | 78.84 | 69.38 |
| FedAvg+FOOGD | **47.87** | **31.16** | **22.17** | **95.24** |
| FedRoD | 57.78 | 30.51 | 68.55 | 73.82 |
| FOSTER | 56.91 | 28.74 | 67.17 | 73.22 |
| FedTHE | 58.23 | 30.24 | 63.63 | 76.83 |
| FedICON | 60.98 | 33.16 | 51.47 | 86.46 |
| FedRoD+FOOGD | **63.27** | **37.26** | **47.26** | **89.31** |

3. **FedRoD** [8] is the personalized federated learning method that adopts two classifiers to achieve both generic and personalized performance.

4. **FOSTER** [83] learns a class-conditional generator to synthesize virtual external-class OOD samples to enhance the detection ability. The weight for the outlier exposure term is set to 0.1.

5. **FedTHE** [26] also contains a personalized classifier. This method adopts an test time adaptation strategy that interpolates the personalized head and global classifier to enforce feature space alignment. We set $\alpha = 0.1$ and $\beta = 0.3$ as suggested in the original paper.

6. **FedICON** [69] performs different contrastive learning during training and test phrase to handle test-time shift problem. Each client finetunes their classifier with learning rate 0.01.

We also compare with centralized OOD generalization and detection methods, adapting them to a FedAvg-like approach for the federated learning scenario.

1. **LogitNorm** [76] applies a straightforward modification to the cross-entropy loss, imposing a constant norm on the logits to improve detection capabilities. $\tau$ is tuned to be set as 0.04.

2. **ATOL** [88] generates OOD data to devise an auxiliary OOD detection task to facilitate real OOD detection. We set the dimension of the latent to be 100, the mean and variance of the Gaussian distribution generating OOD data to be 5.0 and 0.1.

3. **T3A** [25] adjusts a trained linear classifier using a pseudo-prototype. The filter number is set to be 100 for experiments on Cifar10, Cifar100 and TinyImageNet, 50 for experiments on PACS.

# E  Extensive Experiment Results

To summary, we study four additional evaluations: (1) To compare the performance in domain generalization, we also provide the leave-one-out study on PACS [38] in Tab. 11, where FOOGD also obtains better results. (2) In Tab. 10 We compute different detection metrics, i.e., MSP, energy score, and ASH, and validate that Eq. (9) is consistently powerful in detection. (3) We vary the coefficient of SM$^3$D $\lambda_m = \{0.1, 0.2, 0.5, 0.8, 1\}$ in Fig. 11(a)-Fig. 11(b), and vary the coefficient of SAG $\lambda_a = \{0, 0.01, 0.05, 0.1, 0.2, 0.5, 0.8\}$ in Fig. 11(c), to obtain the best modeling in FOOGD. (4) We vary the number of participating clients in Fig. 10 and found FOOGD can have better results among different participating clients.

## E.1  Evaluation on TinyImageNet Dataset.

Additionally, we present the results of TinyImageNet in Tab. 7, and report our main results with variances in Tab. 15 and Tab. 16. It vividly states that FOOGD can also generalize in the task of more classes and more heterogeneous data distribution.

## E.2  Ablation Studies

We devise the variants of FOOGD , i.e., fix backbone, w/o SM$^3$D, and w/o SAG, to study the effectiveness of our three main ideas: (1) obtaining reliable global distribution as guidance, (2) estimating score model by SM$^3$D, and (3) enhancing FL method generalization by SAG, respectively.s From Tab. **??** and its full version in Appendix E.2 Tab. 8 and Tab. 9, simply modeling score model fails in both OOD

Table 8: Cifar10 ablation study on varying $\alpha$ modeled by FedAvg.

| Non-IID | $\alpha = 0.1$ | | | | $\alpha = 0.5$ | | | | $\alpha = 5.0$ | | | |
|---|---|---|---|---|---|---|---|---|---|---|---|---|
| Method | ACC-IN ↑ | ACC-IN-C↑ | FPR95↓ | AUROC↑ | ACC-IN ↑ | ACC-IN-C↑ | FPR95↓ | AUROC↑ | ACC-IN ↑ | ACC-IN-C↑ | FPR95↓ | AUROC↑ |
| fix backbone | 68.03 | 65.44 | 51.27 | 88.49 | 86.59 | 83.72 | 20.40 | 95.82 | 86.50 | 85.08 | 15.44 | 96.96 |
| w/o SM³D | 74.70 | 73.35 | 41.86 | 88.88 | 88.01 | 87.17 | 19.96 | 95.86 | 88.52 | 87.79 | 15.05 | 97.06 |
| w/o SAG | 73.15 | 70.79 | 37.59 | 91.47 | 87.32 | 85.33 | 18.83 | 96.13 | 87.86 | 86.20 | 12.73 | 97.65 |
| FedAvg+FOOGD | **75.09** | **73.71** | **35.32** | **91.21** | **88.36** | **87.26** | **17.78** | **96.53** | **88.90** | **88.25** | **12.02** | **97.77** |

Table 9: Cifar100 ablation study on varying $\alpha$ modeled by FedAvg.

| Non-IID | $\alpha = 0.1$ | | | | $\alpha = 0.5$ | | | | $\alpha = 5.0$ | | | |
|---|---|---|---|---|---|---|---|---|---|---|---|---|
| Method | ACC-IN ↑ | ACC-IN-C↑ | FPR95↓ | AUROC↑ | ACC-IN ↑ | ACC-IN-C↑ | FPR95↓ | AUROC↑ | ACC-IN ↑ | ACC-IN-C↑ | FPR95↓ | AUROC↑ |
| fix backbone | 51.67 | 47.54 | 56.11 | 82.94 | 58.28 | 54.62 | 68.90 | 77.26 | 61.40 | 56.72 | 68.04 | 77.05 |
| w/o SM³D | 53.45 | 51.58 | 43.49 | 89.26 | 61.82 | 59.91 | 62.18 | 84.72 | 64.03 | 62.19 | 64.18 | 83.16 |
| w/o SAG | 53.14 | 48.35 | 37.23 | 91.13 | 60.39 | 55.72 | 60.53 | 85.17 | 62.12 | 57.16 | 59.58 | 82.84 |
| FedAvg+FOOGD | **53.84** | **51.69** | **36.40** | **91.41** | **62.19** | **60.25** | **55.70** | **86.42** | **64.96** | **64.18** | **57.70** | **84.03** |

generalization and detection tasks, since feature extractor not adjusted with the global distribution. When we remove SM³D, the estimation of data probability is severely impacted, bringing no detection capability. On the contrary, the generalization performance decreases once we remove SAG. Moreover, compared with fix backbone, both w/o SM³D and w/o SAG have better generalization and detection results, indicating that it is necessary to introduce the global distribution.

### E.3 Toy Example for validating SM³D

To illustrate the effectiveness of SM³D, we further visualize a density estimation of 2-D toy example in Fig. 3. In detail, we model the red points sampled from target distribution, by tuning a series of coefficients, i.e., $\lambda_m = \{0, 0.05, 0.1, 0.2, 0.4, 0.5, 0.8, 1\}$ in Eq. (8). To start with, the blue generated data contract loosely close to the red target data. Then the distribution divergence gets smaller, making generated distribution overlap with targeted distribution. While, as the effect of MMD increases, the distribution alignment dramatically worsens, even causing the blue generated data to collapse into the expectation of target distribution. As we can see, the mutual impacts between score matching and MMD estimation, SM³D has more compact density estimation when $\lambda_m = 0.1$, compared with blankly using score matching ($\lambda_m = 0$) or simply using MMD ($\lambda_m = 1$). In the brand new objective of density estimation, SM³D expand the searching range and depth of score modeling, making it possible to comprehensively model data density. Moreover, with the calibration of MMD estimation, original data representation and the generated latents based on the score model are effectively matched, bringing more realistic and correct estimation. Hence SM³D could ensure a more aligned and reliable density estimation for sparse and multi-modal data.

### E.4 Detection Score Methods Comparison

To study the effectiveness of our choice, i.e., IsOUT($\cdot$) defined by the norm of score model in Eq. (9), we compare it with existing benchmarks, MSP [20], Energy score [3], and ASH [12]. As listed in Tab. 10, MSP is the runner-up method to detection, and it is flexible to detect in all baseline methods. However, IsOUT($\cdot$) is more competitive and reliable, since it utilizes the global distribution as guidance.

### E.5 Extensive Visualization Results

To explore the wild data distribution of FL OOD methods, we visualize T-SNE of data representations in Fig. 8, and the detection score distributions in Fig. 9, on Cifar10 $\alpha = 5$ for FedAvg, FedRoD,

Table 10: Metric comparison FedRoD+FOOGD on Cifar10.

| Non-IID $\alpha$ | 0.1 | | 0.5 | | 5.0 | |
|---|---|---|---|---|---|---|
| Method | FPR95↓ | AUROC↑ | FPR95↓ | AUROC↑ | FPR95↓ | AUROC↑ |
| MSP | 47.96 | 80.95 | 37.02 | 86.49 | 36.13 | 86.64 |
| Energy | 61.90 | 85.55 | 49.73 | 90.93 | 54.10 | 91.12 |
| ASH | 51.06 | 89.55 | 42.36 | 92.11 | 38.77 | 93.14 |
| +FOOGD | **32.99** | **91.76** | **25.51** | **94.19** | **18.91** | **96.25** |

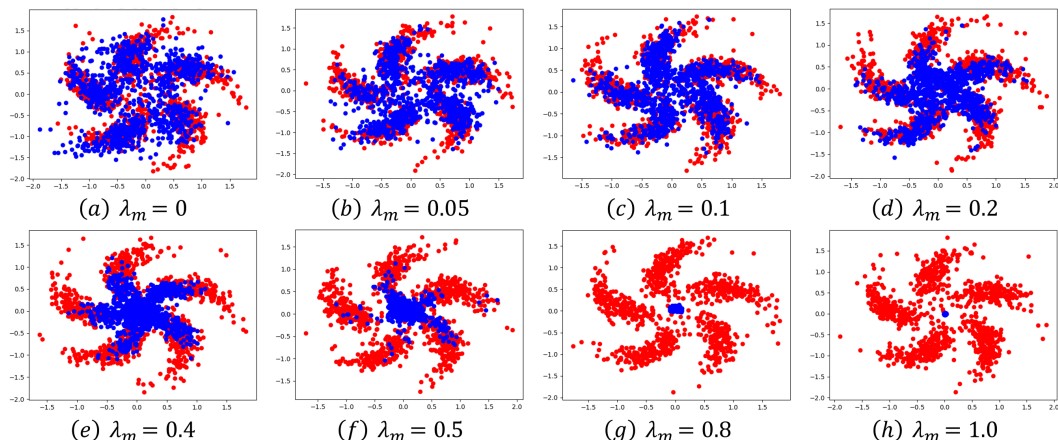

Figure 7: Motivation of SM³D. The red points are sampled from target distribution, while the blue points are generated via Langevin dynamic sampling from random noise.

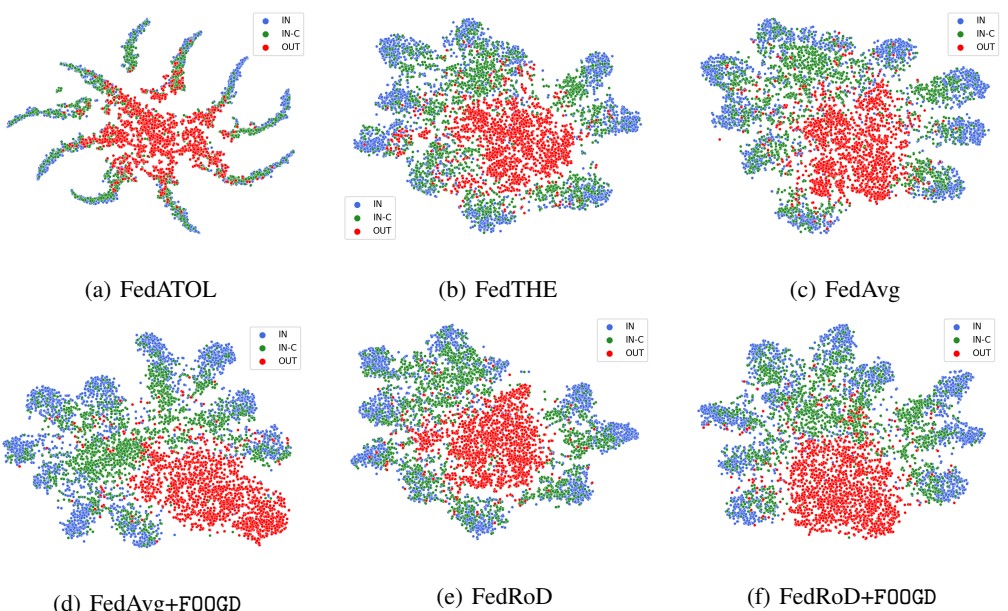

Figure 8: T-SNE visualizations of FedAvg and FedRoD with FOOGD.

FedAvg+FOOGD , FedRoD+FOOGD and their runner-up methods, FedATOL and FedTHE, respectively. It is evident that FOOGD represents IN-C data more tight with IN data, and constructs a comparably clear decision boundary between IN data and OUT data. Besides, we also discover that FOOGD will push OUT data away from its IN and IN-C data, which validates the guidance from the global distribution. Additionally, in Fig. 9, FOOGD makes the modes among IN, IN-C, and OUT, more separable than existing methods. This also proves the effectiveness of FOOGD in detection task.

### E.6 Client Generalization on PACS Dataset

To validate the effectiveness of FOOGD in domain generalization tasks, i.e., each client contains one domain data and we train domain generalization model by leave-one-out, following FedIIR [17]. To obtain a fair comparison, we pretrain all models from scratch and utilize adaption methods as stated in their main paper, instead of using a public ImageNet pre-trained model. In terms of Tab. 11, FOOGD obtains performance improvements for FedAvg and FedRoD. Compared with existing adaption methods, FOOGD achieves outstanding results even in the toughest task, i.e., leaving Sketch domain

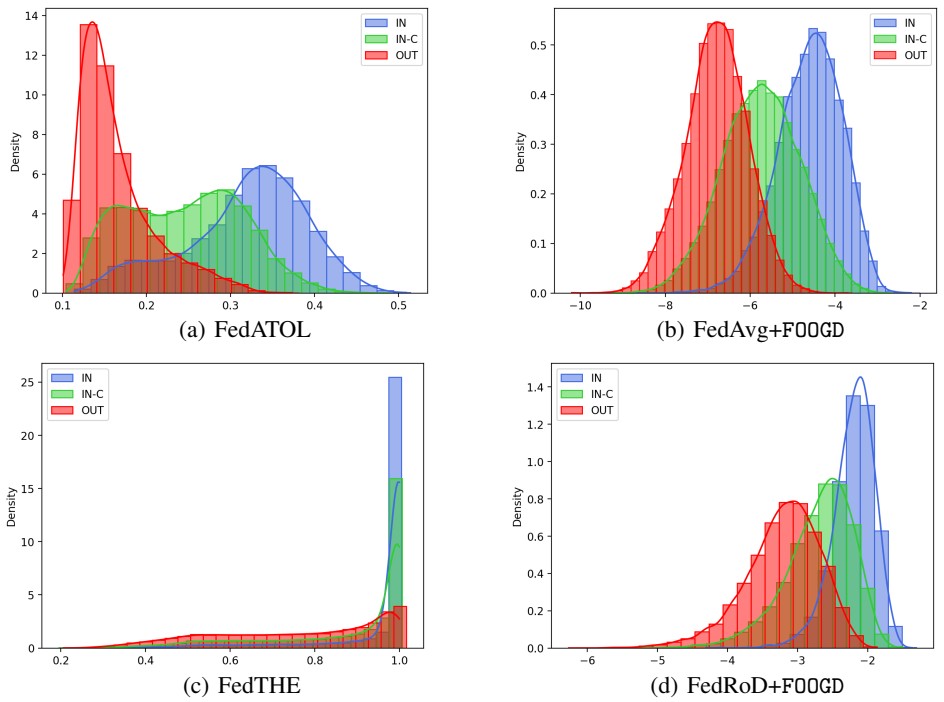

Figure 9: Detection score distribution of FedAvg and FedRoD with FOOGD on Cifar10 ($\alpha = 5$).

Table 11: OOD generalization task for PACS.

| Method \Domain | Art Painting | Cartoon | Photo | Sketch | Average |
|---|---|---|---|---|---|
| FedAvg | 97.21 | 62.58 | 91.00 | 35.28 | 71.52 |
| FedRoD | 93.45 | 88.85 | 89.34 | 29.95 | 75.39 |
| FedT3A | 97.13 | 75.71 | 93.21 | 37.40 | 75.86 |
| FedIIR | 86.86 | 80.29 | 88.98 | 31.38 | 71.88 |
| FedTHE | 96.17 | 90.72 | 93.57 | 29.14 | 77.40 |
| FedICON | 50.42 | 53.36 | 52.19 | 50.87 | 51.71 |
| FedAvg+FOOGD | **97.46** | **89.32** | **91.48** | 41.40 | **79.92** |
| FedRoD+FOOGD | **97.85** | **92.31** | **93.01** | **50.95** | **83.53** |

out. This also concludes that FOOGD is capable of inter-client generalization, since FOOGD has utilized global distribution knowledge.

### E.7 Extensive Experiments on Other IN-C and OUT data

In this part, we study the performance evaluation of FOOGD in additional IN-C and OUT datasets. In Tab. 12, we can find that FOOGD consistently enhances the detection capability for different OUT data, validating for the effectiveness of estimating global distribution via $SM^3D$. Meanwhile, we compute the average results of different IN-C data on Fig. 12 and provide the details in Tab. 13 and Tab. 14. FOOGD consistently improve the generalization in all unseen IN-C data, indicating the effectiveness of enhancing feature extractor via SAG.

### E.8 The Study of Hyper-parameter Sensitivity

We vary the coefficient of $SM^3D$ $\lambda_m = \{0.1, 0.2, 0.5, 0.8, 1\}$ in Fig. 11(a)-Fig. 11(b), and vary the coefficient of SAG $\lambda_a = \{0, 0.01, 0.05, 0.1, 0.2, 0.5, 0.8\}$ in Fig. 11(c), to obtain the best modeling in FOOGD. To study the effect of different client numbers, we vary the number of participating clients $K = \{5, 10, 20, 50\}$ in Fig. 10 and find FOOGD can have better results among different participating clients.

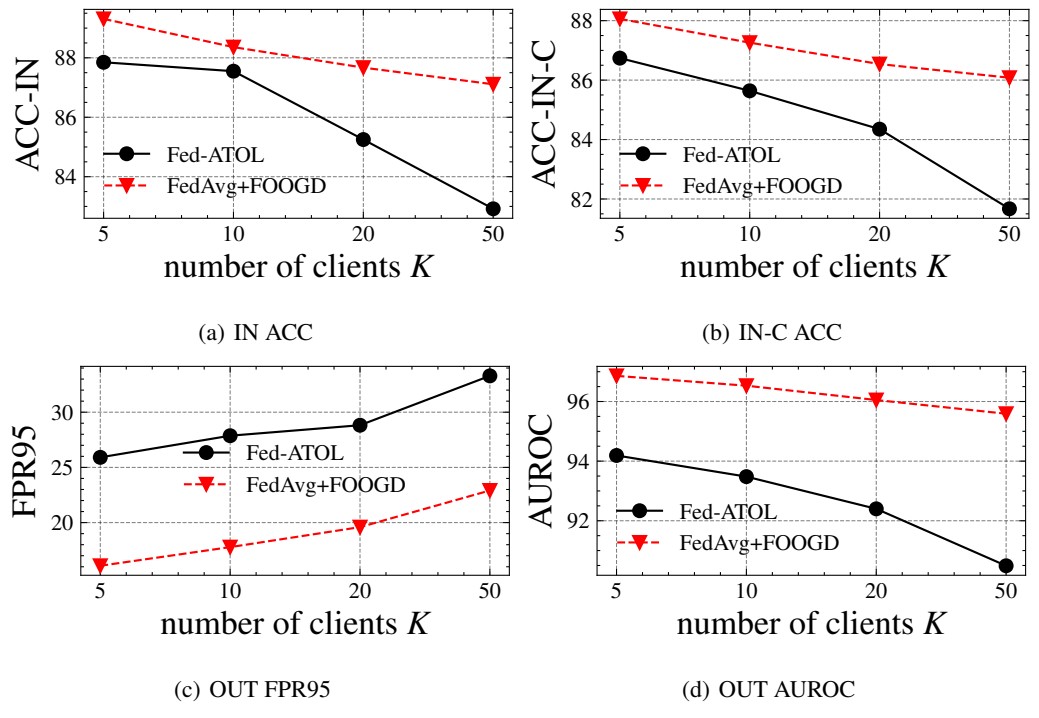

(a) IN ACC

(b) IN-C ACC

(c) OUT FPR95

(d) OUT AUROC

Figure 10: Effect of participating clients numbers $K$.

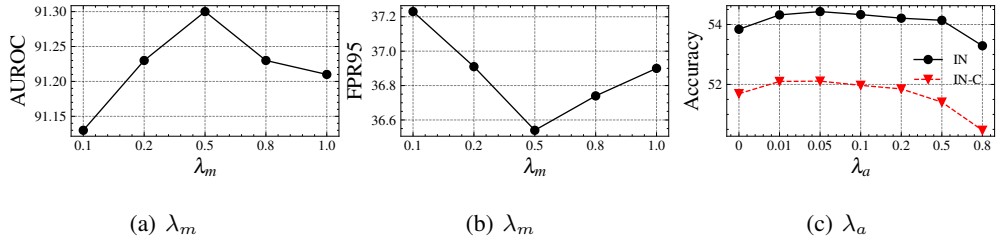

(a) $\lambda_m$

(b) $\lambda_m$

(c) $\lambda_a$

Figure 11: Effect of $\lambda_m$ and $\lambda_a$.

Table 12: Other detection results on Cifar10 ($\alpha = 0.1$).

| OUT Data | iSUN | | SVHN | | LSUN-R | | Texture | |
|---|---|---|---|---|---|---|---|---|
| Method | FPR95↓ | AUROC↑ | FPR95↓ | AUROC↑ | FPR95↓ | AUROC↑ | FPR95↓ | AUROC↑ |
| FedAvg | 62.10 | 76.29 | 80.02 | 62.14 | 62.01 | 77.02 | 80.53 | 66.23 |
| FedLN | 66.41 | 76.03 | 70.95 | 76.82 | 61.31 | 78.34 | 93.90 | 71.99 |
| FedATOL | 61.01 | 80.05 | 85.39 | 82.17 | 64.01 | 79.89 | 66.33 | 78.77 |
| FedIIR | 57.86 | 77.98 | 83.68 | 64.04 | 58.44 | 78.69 | 91.72 | 62.32 |
| FedAvg +FOOGD | **37.55** | **91.22** | **44.59** | **87.63** | **44.16** | **90.16** | **28.60** | **91.75** |
| FedRoD | 43.40 | 82.83 | 40.72 | 83.55 | 41.80 | 82.92 | 53.24 | 81.52 |
| FOSTER | 48.73 | 76.29 | 39.55 | 83.07 | 48.09 | 76.24 | 54.23 | 77.62 |
| FedTHE | 43.72 | 83.50 | 39.22 | 85.95 | 42.95 | 83.46 | 53.58 | 82.19 |
| FedICON | 49.98 | 82.95 | 34.94 | 85.56 | 49.05 | 83.30 | 51.57 | 80.96 |
| FedRoD +FOOGD | **36.17** | **88.69** | **17.61** | **94.56** | **41.46** | **92.80** | **19.46** | **93.39** |

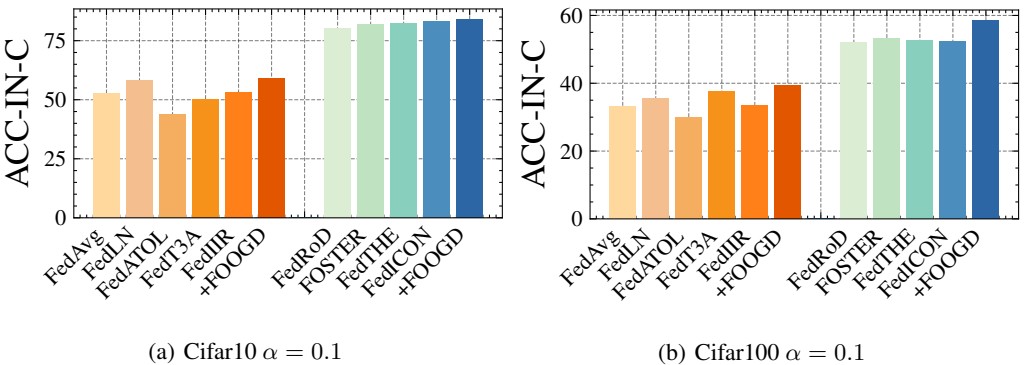

(a) Cifar10 $\alpha = 0.1$  (b) Cifar100 $\alpha = 0.1$

Figure 12: Average generalization results. +FOOGD is short for FedAvg+FOOGD and FedRoD+FOOGD, respectively.

Table 13: IN-C generalization for FL methods trained on Cifar10($\alpha = 0.1$).

| IN-C Type | FedAvg | FedLN | Fed-ATOL | Fed-T3A | FedIIR | FedAvg+FOOGD | FedRoD | FOSTER | FedTHE | FedICON | FedRoD+FOOGD |
|---|---|---|---|---|---|---|---|---|---|---|---|
| Brightness | 65.73 | 71.77 | 54.44 | 61.14 | 66.12 | **73.71** | 89.90 | 88.70 | 89.61 | 89.18 | **92.74** |
| Fog | 53.89 | 60.82 | 48.17 | 49.52 | 54.85 | **60.96** | 81.48 | 83.35 | 83.85 | 86.35 | **86.12** |
| Frosted glass blur | 43.13 | 42.33 | 28.97 | 40.41 | **44.53** | 45.80 | 68.49 | 75.03 | 75.42 | 70.16 | **75.44** |
| Motion blur | 41.30 | **52.65** | 41.52 | 45.51 | 44.23 | 51.05 | 77.86 | 81.76 | 80.10 | 81.22 | **81.98** |
| Snow | 54.80 | 60.55 | 45.64 | 51.52 | 55.52 | **61.90** | 81.11 | 83.05 | 83.43 | 82.32 | **86.38** |
| Contrast | 41.25 | 45.02 | 38.90 | 36.93 | 41.35 | **49.14** | 71.87 | 72.82 | 71.85 | **87.09** | 76.84 |
| Frost | 56.21 | 58.22 | 41.25 | 52.16 | 55.91 | **63.84** | 81.20 | 82.76 | 82.31 | 83.04 | **86.62** |
| Impulse noise W | 49.32 | 50.52 | 40.36 | 42.79 | 48.45 | **52.33** | 75.25 | 76.52 | 78.49 | 76.43 | **80.45** |
| Pixelate | 56.88 | 62.00 | 46.20 | 53.34 | 59.10 | **64.37** | 85.65 | 85.71 | 84.74 | 85.41 | **88.08** |
| Defouce blur | 52.37 | **61.08** | 46.36 | 52.60 | 52.72 | 58.66 | 82.13 | **84.44** | 84.35 | 86.63 | 83.38 |
| Jpeg | 61.56 | **68.61** | 47.94 | 57.64 | 60.46 | 66.55 | 86.61 | 86.68 | 87.50 | 86.37 | **89.80** |
| Elastic transform | 52.12 | **61.29** | 45.35 | 52.45 | 53.21 | 59.18 | 83.37 | 84.82 | 84.67 | 83.48 | **86.27** |
| Gaussian Noise | 48.66 | 50.25 | 35.20 | 45.27 | 49.15 | **53.92** | 73.75 | 76.80 | 78.37 | 76.82 | **81.55** |
| Shot noise | 52.73 | 54.55 | 39.57 | 48.82 | 53.09 | **58.31** | 76.62 | 78.94 | 80.31 | 80.34 | **83.12** |
| Zoom blur | 45.15 | **54.88** | 42.33 | 49.48 | 46.57 | 52.97 | 79.27 | 82.41 | 82.07 | **86.05** | 80.88 |
| Spatter | 62.18 | **67.33** | 51.54 | 55.25 | 60.97 | 65.31 | 85.55 | 85.63 | 87.66 | 85.23 | **87.90** |
| Gaussian blur | 46.86 | **55.64** | 43.23 | 48.52 | 47.51 | 53.26 | 77.97 | 81.88 | 81.32 | **85.08** | 79.16 |
| Saturate | 63.62 | 71.76 | 54.39 | 58.41 | 63.32 | **71.98** | 88.49 | 87.06 | 88.73 | 88.64 | **91.81** |
| Speckle noise | 52.25 | 54.30 | 40.03 | 48.38 | 53.20 | **57.72** | 76.60 | 78.63 | 79.72 | 80.43 | **81.83** |
| average | 52.63 | 58.08 | 43.76 | 50.01 | 53.17 | **59.00** | 80.17 | 81.95 | 82.34 | 83.17 | **84.23** |

Table 14: IN-C generalization for FL methods trained on Cifar100($\alpha = 0.1$).

| IN-C Type | FedAvg | FedLN | FedATOL | FedT3A | FedIIR | FedAvg+F00GD | FedRoD | FOSTER | FedTHE | FedICON | FedRoD+F00GD |
|---|---|---|---|---|---|---|---|---|---|---|---|
| Brightness | 46.85 | 48.15 | 41.08 | 51.50 | 47.88 | **51.69** | 69.26 | 67.50 | 69.09 | 67.79 | **75.70** |
| Fog | 36.15 | 37.11 | 33.87 | **41.45** | 36.80 | 40.98 | 56.26 | 56.01 | 57.69 | 56.16 | **64.23** |
| Frosted glass blur | 20.96 | 27.32 | 18.40 | 27.48 | 19.67 | **27.44** | 34.86 | **39.94** | 36.78 | 37.12 | 38.83 |
| Motion blur | 32.95 | 35.09 | 30.23 | 37.65 | 33.34 | **39.68** | 54.24 | 53.53 | 53.13 | 53.58 | **60.09** |
| Snow | 35.09 | 38.60 | 32.39 | 39.41 | 35.69 | **40.64** | 54.61 | 55.40 | 55.80 | 54.79 | **61.34** |
| Contrast | 26.39 | 27.10 | 26.96 | 29.88 | 26.94 | **30.98** | 42.32 | 42.91 | 44.98 | 43.62 | **51.06** |
| Frost | 32.53 | 35.38 | 29.50 | 37.40 | 33.33 | **38.54** | 50.08 | 52.36 | 53.15 | 51.08 | **60.58** |
| Impulse noise W | 22.99 | 24.26 | 21.58 | 23.65 | 21.84 | **26.24** | 38.66 | 40.68 | 38.44 | 39.21 | **43.30** |
| Pixelate | 34.41 | 36.11 | 32.51 | 42.10 | 33.31 | **42.52** | 53.73 | 54.97 | 51.60 | 52.17 | **62.88** |
| Defouce blur | 39.17 | 41.05 | 34.67 | 44.18 | 39.92 | **46.23** | 61.02 | 60.76 | 60.75 | 60.08 | **64.20** |
| Jpeg | 41.17 | 43.36 | 33.64 | **46.63** | 41.90 | 45.81 | 62.51 | 63.10 | 63.55 | 62.57 | **65.51** |
| Elastic transform | 38.65 | 41.49 | 33.89 | 44.72 | 39.36 | **47.47** | 61.36 | 61.11 | 61.34 | 60.13 | **65.66** |
| Gaussian Noise | 21.21 | 24.83 | 19.67 | 22.50 | 21.79 | **28.28** | 33.41 | 38.43 | 35.51 | 37.00 | **46.47** |
| Shot noise | 26.37 | 30.28 | 24.01 | 29.20 | 27.03 | **32.81** | 40.32 | 44.96 | 42.33 | 43.64 | **53.00** |
| Zoom blur | 33.82 | 36.51 | 30.63 | 39.34 | 34.75 | **41.62** | 56.77 | 56.06 | 55.92 | 55.38 | **60.09** |
| Spatter | 42.41 | 43.90 | 36.39 | 47.32 | 42.04 | **49.59** | 63.20 | 63.32 | **64.11** | 62.92 | 63.86 |
| Gaussian blur | 34.18 | 36.29 | 30.63 | 38.98 | 35.39 | **40.63** | 55.11 | 55.41 | 54.56 | 54.41 | **58.32** |
| Saturate | 38.59 | 39.43 | 35.61 | 42.19 | 38.92 | **44.87** | 59.39 | 58.40 | 59.87 | 58.24 | **66.93** |
| Speckle noise | 26.43 | 30.53 | 24.47 | 29.31 | 27.47 | **32.86** | 41.75 | 45.67 | 43.39 | 44.28 | **52.63** |
| Average | 33.17 | 35.62 | 30.01 | 37.63 | 33.55 | **39.42** | 52.05 | 53.19 | 52.74 | 52.32 | **58.67** |

Table 15: Main results of federated OOD detection and generalization on Cifar10. We report the ACC of brightness as IN-C ACC, the FPR95 and AUROC of LSUN-C as OUT performance.

| Non-IID | α = 0.1 | | | | α = 0.5 | | | | α = 5.0 | | | |
|---|---|---|---|---|---|---|---|---|---|---|---|---|
| Method | ACC-IN↑ | ACC-IN-C↑ | FPR95↓ | AUROC↑ | ACC-IN↑ | ACC-IN-C↑ | FPR95↓ | AUROC↑ | ACC-IN↑ | ACC-IN-C↑ | FPR95↓ | AUROC↑ |
| FedAvg | 68.03 ± 1.17 | 65.44 ± 1.18 | 83.41 ± 1.57 | 58.05 ± 0.89 | 86.59 ± 1.13 | 83.72 ± 1.74 | 43.70 ± 0.83 | 84.18 ± 0.23 | 86.50 ± 0.33 | 85.08 ± 0.49 | 38.24 ± 0.55 | 85.37 ± 0.29 |
| FedLN | 75.24 ± 0.44 | 71.77 ± 0.67 | 56.14 ± 0.91 | 84.14 ± 0.37 | 86.10 ± 0.89 | 84.20 ± 1.82 | 39.26 ± 1.14 | 89.64 ± 0.52 | 87.20 ± 1.26 | 85.08 ± 1.43 | 33.33 ± 2.38 | 90.87 ± 0.58 |
| FedATOL | 55.93 ± 1.87 | 54.44 ± 1.72 | 49.50 ± 1.59 | 86.22 ± 2.74 | 87.55 ± 0.91 | 85.64 ± 0.54 | 27.87 ± 1.32 | 93.48 ± 0.69 | 89.27 ± 0.68 | 88.28 ± 1.32 | 19.66 ± 2.62 | 95.25 ± 0.78 |
| FedT3A | 68.03 ± 1.17 | 61.52 ± 1.39 | 78.12 ± 1.57 | 63.64 ± 1.38 | 86.59 ± 1.13 | 82.85 ± 0.44 | 43.70 ± 0.83 | 84.18 ± 0.23 | 86.50 ± 0.33 | 85.01 ± 1.46 | 38.24 ± 0.55 | 85.37 ± 0.29 |
| FedIIR | 68.26 ± 0.66 | 66.12 ± 0.74 | 79.48 ± 0.99 | 63.31 ± 1.38 | 86.75 ± 0.98 | 84.75 ± 1.92 | 40.91 ± 0.64 | 84.94 ± 0.49 | 87.77 ± 0.66 | 86.10 ± 0.95 | 34.69 ± 1.07 | 87.66 ± 0.47 |
| FedAvg+FOOGD | **75.09 ± 0.79** | **73.71 ± 0.93** | **35.32 ± 1.02** | **91.21 ± 0.78** | **88.36 ± 0.43** | **87.26 ± 0.86** | **17.78 ± 0.62** | **96.53 ± 0.18** | **88.90 ± 0.29** | **88.25 ± 0.12** | **12.02 ± 0.34** | **97.77 ± 0.41** |
| FedRoD | 91.15 ± 0.87 | 89.90 ± 0.85 | 47.97 ± 1.88 | 80.96 ± 0.90 | 89.62 ± 0.55 | 87.70 ± 0.80 | 37.03 ± 1.40 | 86.50 ± 0.97 | 87.69 ± 0.88 | 86.26 ± 1.19 | 36.13 ± 1.12 | 86.65 ± 0.36 |
| FOSTER | 90.22 ± 0.88 | 88.70 ± 0.82 | 47.40 ± 1.27 | 77.43 ± 0.93 | 86.92 ± 1.85 | 85.82 ± 1.10 | 42.03 ± 1.51 | 83.91 ± 1.11 | 87.83 ± 1.38 | 85.96 ± 1.02 | 36.42 ± 1.14 | 86.19 ± 0.87 |
| FedTHE | 91.05 ± 0.66 | 89.71 ± 0.91 | 58.14 ± 2.79 | 82.04 ± 1.15 | 89.14 ± 0.93 | 87.68 ± 0.41 | 40.28 ± 2.43 | 85.30 ± 1.91 | 88.14 ± 0.24 | 86.18 ± 0.57 | 35.35 ± 1.94 | 86.79 ± 0.37 |
| FedICON | 89.06 ± 0.43 | 89.18 ± 0.81 | 48.22 ± 1.48 | 81.28 ± 0.44 | 75.83 ± 1.07 | 75.35 ± 0.36 | 56.19 ± 1.58 | 79.88 ± 0.51 | 87.20 ± 1.13 | 85.39 ± 0.99 | 35.63 ± 1.16 | 86.45 ± 0.41 |
| FedRoD+FOOGD | **93.51 ± 0.65** | **92.74 ± 0.46** | **32.99 ± 1.30** | **91.76 ± 0.26** | **90.46 ± 0.78** | **90.16 ± 0.51** | **25.51 ± 1.46** | **94.19 ± 0.78** | **89.44 ± 0.88** | **88.62 ± 0.37** | **18.91 ± 0.96** | **96.25 ± 0.22** |

Table 16: Main results of federated OOD detection and generalization on Cifar100. We report the ACC of brightness as IN-C ACC, the FPR95 and AUROC of LSUN-C as OUT performance.

| Non-IID | α = 0.1 | | | | α = 0.5 | | | | α = 5.0 | | | |
|---|---|---|---|---|---|---|---|---|---|---|---|---|
| Method | ACC-IN↑ | ACC-IN-C↑ | FPR95↓ | AUROC↑ | ACC-IN↑ | ACC-IN-C↑ | FPR95↓ | AUROC↑ | ACC-IN↑ | ACC-IN-C↑ | FPR95↓ | AUROC↑ |
| FedAvg | 51.67 ± 1.37 | 47.54 ± 0.48 | 78.35 ± 1.64 | 67.16 ± 1.17 | 58.28 ± 0.48 | 54.62 ± 0.67 | 72.84 ± 0.81 | 70.86 ± 1.52 | 61.40 ± 0.12 | 56.72 ± 0.17 | 72.68 ± 0.34 | 70.59 ± 0.19 |
| FedLN | 52.48 ± 1.41 | 48.15 ± 1.57 | 66.94 ± 1.61 | 74.82 ± 0.50 | 59.39 ± 0.72 | 53.86 ± 1.23 | 68.31 ± 1.24 | 73.41 ± 0.33 | 61.00 ± 0.40 | 56.33 ± 0.82 | 69.18 ± 0.46 | 75.87 ± 0.74 |
| FedATOL | 43.65 ± 0.54 | 41.08 ± 0.60 | 65.26 ± 0.96 | 81.64 ± 0.33 | 60.62 ± 0.61 | 56.63 ± 0.91 | 70.10 ± 0.81 | 79.27 ± 0.61 | 64.16 ± 0.81 | 63.61 ± 0.42 | 80.27 ± 1.61 | 60.51 ± 1.75 |
| FedT3A | 51.67 ± 1.37 | 51.50 ± 0.49 | 78.35 ± 1.64 | 67.16 ± 1.17 | 58.28 ± 0.48 | 55.42 ± 1.63 | 72.84 ± 1.56 | 70.86 ± 1.52 | 61.40 ± 0.12 | 55.51 ± 0.96 | 72.68 ± 0.34 | 70.59 ± 0.19 |
| FedIIR | 51.63 ± 0.61 | 47.88 ± 1.19 | 81.91 ± 0.47 | 63.99 ± 0.53 | 58.66 ± 0.41 | 55.72 ± 0.29 | 77.62 ± 1.10 | 65.87 ± 0.46 | 61.70 ± 0.76 | 57.65 ± 0.80 | 72.57 ± 0.37 | 69.07 ± 0.52 |
| FedAvg+FOOGD | 53.84 ± 0.83 | 51.69 ± 0.12 | 36.40 ± 1.11 | 91.41 ± 0.36 | 61.82 ± 0.20 | 59.91 ± 0.31 | 55.70 ± 0.78 | 86.42 ± 0.24 | 64.96 ± 0.51 | 64.18 ± 0.31 | 57.70 ± 0.87 | 84.03 ± 0.15 |
| FedRoD | 73.13 ± 0.85 | 69.26 ± 0.41 | 66.34 ± 1.53 | 73.02 ± 1.82 | 66.88 ± 0.61 | 61.28 ± 0.98 | 70.13 ± 0.86 | 69.48 ± 0.65 | 61.34 ± 0.78 | 55.80 ± 1.21 | 74.86 ± 0.98 | 67.76 ± 1.31 |
| FOSTER | 72.54 ± 1.51 | 67.50 ± 0.57 | 61.25 ± 1.05 | 75.44 ± 0.89 | 62.45 ± 0.55 | 57.62 ± 0.87 | 73.26 ± 1.13 | 68.71 ± 0.85 | 53.80 ± 0.31 | 49.28 ± 0.74 | 76.94 ± 1.62 | 65.47 ± 1.72 |
| FedTHE | 73.83 ± 0.48 | 69.09 ± 0.56 | 64.73 ± 0.79 | 75.16 ± 0.34 | 66.22 ± 0.68 | 61.19 ± 0.92 | 72.95 ± 1.84 | 69.38 ± 1.64 | 61.03 ± 0.22 | 57.03 ± 0.16 | 71.43 ± 0.64 | 69.01 ± 0.87 |
| FedICON | 72.22 ± 0.72 | 67.79 ± 0.31 | 61.36 ± 0.39 | 77.12 ± 0.55 | 65.86 ± 0.81 | 61.83 ± 0.55 | 69.99 ± 1.02 | 71.03 ± 0.39 | 62.11 ± 0.74 | 57.62 ± 0.28 | 70.91 ± 0.97 | 70.84 ± 0.73 |
| FedRoD+FOOGD | 77.88 ± 0.28 | 75.70 ± 0.26 | 58.81 ± 0.48 | 86.07 ± 0.39 | 70.30 ± 0.46 | 68.23 ± 0.25 | 45.19 ± 0.67 | 89.59 ± 0.28 | 64.94 ± 0.79 | 62.56 ± 0.72 | 65.18 ± 1.19 | 80.47 ± 0.32 |

