# OpenReview forum: "FOOGD: Federated Collaboration for Both Out-of-distribution Generalization and Detection"
_NeurIPS.cc/2024/Conference — NeurIPS 2024 poster_

### Official Review · Reviewer_dEK5 · 2024-06-15

**Soundness:** 3
**Presentation:** 2
**Contribution:** 2
**Rating:** 5
**Confidence:** 4

**Summary:**

This paper introduces a FL framework FOOGD aimed at addressing the simultaneous challenges of the out-of-distribution (OOD) generalization and OOD detection in decentralized environments. FOOGD estimates the data distribution of different clients through SM3D and introduces SAG to ensure the consistency of features.

**Strengths:**

FOOGD is designed to handle both OOD generalization and OOD detection tasks and integrates mechanisms to manage covariate shifts and semantic shifts within the FL setup, which is helpful for deployment environments. The use of SM3D allows for accurate estimation of the probability density of client data distributions and is used to identify out-of-distribution samples. In addition, SAG helps maintain feature space invariance between in-distribution and augmented data samples. Finally, this paper demonstrates the effectiveness of FOOGD through experiments.

**Weaknesses:**

There is  no training-log provided in supp materials. There are also questions regarding the foundation methods as listed below.

**Questions:**

(1) The dual approach of handling both OOD generalization and detection introduces significant computational complexity and overhead. However, FOOGD is not clearly stated, and there are no experiments to quantitatively analyze it. The iterative process of updating and aggregating both main task models and score models can still be costly in terms of bandwidth and synchronization requirements among clients.
(2) The need to transmit detailed model parameters and score function values between clients and the server can lead to high communication costs. In addition, the score function values may cause privacy leakage issues, thereby attacking the score model.
(3) While the FOOGD framework aims to generalize across different client distributions, the inherent diversity and possible sparsity of client data can cause the model to overly adapt to specific local patterns, reducing its effectiveness in capturing broader, global trends.
(4) The effectiveness of FOOGD heavily relies on the accuracy of the local density estimation provided by the SM3D component. However, in scenarios where data distributions are highly complex or sparse, it may lead to less reliable detection of OOD samples and potential false positives or negatives.

**Limitations:**

The authors indicated the paper's focus (on privacy issues) as an limitation but not elaborated in paper itself.

---

> ### Author Rebuttal · Authors · 2024-08-04
>
> Thanks for your professional and detailed review!
>
>  W: The training log is not provided.
>
> A: To validate the effectiveness and implementation, we release a well-trained FedAVG-FOOGD model on Cifar10 $\alpha=0.5$ in supplemental materials. And we are willing to release our project as well as the training logs in our later version.
>
>  Q1: The computational and communication analysis of introducing score models.
>
> A1: **FOOGD introduces fairly controllable communication cost and computational cost, but brings competitive and flexible adaptation capability in OOD shifts data.**
> **(1) computation cost: the complexity of $SM^3D$ is $O((T+1)BD + B^2D)$, and the complexity of SAG is $O(B^2 D)$, where $B$ is the batch size of samples,$T$ is sampling steps, and $D$ is feature dimension.**
> $SM^3D$ Module consists of Score Matching ($O(BD)$) Langevin Dynamic Sampling ($O(TBD)$), and Maximum Mean Discrepancy computing ($O(B^2D)$), bringing overall computation as $O((T+1)BD + B^2D)$.
> SAG Module needs to compute Kernelized Stein Discrepancy, requiring $O(B^2D)$.
>
> We also provide the empirical reports in Tab. a-b. While both $SM^3D$ and SAG introduce extra computation burden, the most proportional of computation is due to main task inference, e.g., about 89% for training $SM^3D$ and 95% for SAG.Both SM3D and SAG add extra computation, but the main task remains the most significant part.
>
>  **Table a. Average computation (s) of batch samples for training score model.**
> || Main task|$SM^3D$|Total|
> |-|-|-|-|
> |Cifar10|0.0252| 0.0031|0.0283|
> |Cifar100|0.0281| 0.0031|0.0312|
>
>
>   **Table b. Average computation (s) of batch samples for training feature extractor.**
> |  | Main task | SAG | Total |
> |-|-|-|-|
> |Cifar10|0.0503|0.00212|0.0524|
> |Cifar100|0.0561|0.00236|0.0585|
>
>  **(2) communication cost: we transmit maintask model (2,243,546 parameters) and score model (132,992 parameters). Score model brings additional but controllable communication burden, increasing 5.93% overburden.** We can observe that the most significant communication burden is derived from main task rather than additional modules.
>
> Q2: Introduce score model bring communication cost and privacy leackage.
> A2:**Score model introduce slightly controllable communication burden, and no additional privacy risk.(1) Communication Burden:** FOOGD only transmits score model parameters between clients and the server, adding just 5.93% extra overhead. Score function values are not transmitted; the score models carry only distribution knowledge. The communication cost details are provided in A1.**(2) Privacy:** FOOGD poses no additional privacy risk compared to existing FL methods. The score model captures data probability in latent distribution, revealing only the direction and distance towards the in-distribution, making it hard to reverse-engineer original data. The only risk exposure of each client is transmiting model parameters, i.e., feature extractor model and score model. But this kind of privacy exposure can be mitigated via enhanced privacy-preserving techniques, e.g., differential privacy. We will explore this in our future study.
>
>  Q3: The inherent diversity and possible sparsity of client data can cause the model to overly adapt to specific local patterns, reducing its effectiveness in capturing broader, global trends.
> A3: **FOOGD avoids over-adapting to local patterns and effectively captures global distributions, even with diverse and sparse client data.**
> (1) **Local Modeling:** SM3D estimates data distributions broadly, addressing sparsity, and uses MMD for accurate alignment. SAG ensures invariant alignment, enhancing generalization.
> (2) **Global Modeling:** Aggregated local score models form a global score model, ensuring consistent global distribution in SAG, as guaranteed by theorem 4.1.
> (3) **Empirical Evaluation:** For α=0.1, FOOGD outperforms baselines on all datasets, e.g., Cifar10, TinyImageNet and PACS, demonstrating robust performance with diverse and sparse data.
> These points highlight FOOGD's superior handling of diverse and sparse data in federated learning.
>
>  Q4:  When data distributions are highly complex or sparse, it may lead to less reliable detection of OOD samples and potential false positives or negatives.
>  A4: **$SM^3D$ accurately captures local data density for complex and sparse data, providing reliable OOD detection.**
>  **Firstly, we explain the mechanism of $SM^3D$.**
>   As introduced in Sec. 3.2, $SM^3D$ achieves better distribution modeling for complex and sparse data, since it benefits from both denoising score matching (DSM) and maximum mean discrepancy (MMD). $SM^3D$ takes the approximate distribution prior from DSM, and utilizes MMD to mitigate the impact of complex and sparse data pattern by minimizing the distribution between generative data samples from wider distribution space, and original data samples.
> This is validated in lines 184-201. The visualizations show that with a balanced trade-off $\lambda_m=0.1$), $SM^3D$ achieves more compact density estimation, providing comprehensive data density estimation.
> **Secondly, we also provide various empirical studies that validate the effectiveness of $SM^3D$.**
> **(1) $SM^3D$ directly enhances federated OOD detection even in sparse and complex data.** For instance, in the Cifar10 experiments with α = 0.1 Tab. 1-2, FOOGD with $SM^3D$ outperforms other baseline methods. Besides,  $SM^3D$ creates more separable modes among IN, IN-C, and OUT data, which is good for detection.
> **(2) Removing $SM^3D$ causes significant performance drop.** Removing $SM^3D$, the AUROC drops to 88.88%, indicating the necessity of $SM^3D$ for optimal performance.
> **(3) $SM^3D$ implicitly improves federated OOD generalization, even for unseen and complex data.** With $SM^3D$, FOOGD has  an average accuracy of 79.92% on PACS. **These experiments collectively demonstrate the effectiveness of $SM^3D$ in federated OOD detection, and generalization.**

---

> > ### Comment · Reviewer_dEK5 · 2024-08-08
> >
> > It clears my concerns and I have lifted scores.

---

> > > ### Author Response · Authors · 2024-08-11
> > > **Response to Reviwer dEK5**
> > >
> > > Thank you very much for your reviews and response. We sincerely appreciate it for your time and effort in evaluating our work. We are grateful for your valuable feedback and are particularly thankful for your recognition of the contributions we have made.

---

### Official Review · Reviewer_cFix · 2024-07-08

**Soundness:** 3
**Presentation:** 3
**Contribution:** 3
**Rating:** 5
**Confidence:** 4

**Summary:**

This paper introduces FOOGD, a federated collaboration framework designed to achieve both out-of-distribution (OOD) generalization and detection. FOOGD estimates the probability density of each client to obtain a reliable global distribution and incorporates the SM3D model and SAG module to enhance the detection capability of the score model and the generalization ability of the feature extractor.

**Strengths:**

1.	FOOGD is the first framework in federated learning to simultaneously address OOD generalization and detection, demonstrating significant innovation.
2.	The paper is written clearly and coherently, and it is generally well-organized.
3.	The authors have made an effort to highlight their contributions, providing a reasonably good overview of the key points and innovations presented in the paper.

**Weaknesses:**

1.	The experimental section discusses the results and performance comparisons. Although this section provides performance metrics for multiple datasets, the description of the chosen statistical analysis methods (such as whether hypothesis testing or confidence interval calculation was performed) is insufficient, affecting the reliability of the result interpretation.
2.	Evaluation metrics like ACC-IN and ACC-IN-C are not explained.
3.	The related work section only introduces some comparison methods and lacks more related work, such as FedAvg. Besides, some references are incomplete, such as ref. 46.
4.	No open-source code has been provided.

**Questions:**

1.	Could the experimental results section provide a more detailed analysis of OOD to better explain the superiority of the FOOGD method in this aspect?

**Limitations:**

See the above weakness.

---

> ### Author Rebuttal · Authors · 2024-08-04
>
> We appreciate it a lot for your insightful reviews.
>
> W1: Although this section provides performance metrics for multiple datasets, the description of the chosen statistical analysis methods (such as whether hypothesis testing or confidence interval calculation was performed) is insufficient, affecting the reliability of the result interpretation.
>
> A1: We follow FedTHE to choose accuracy for OOD generalization, and SCONE to choose AUROC and FPR95 for OOD detection, for fair comparison.  We will extend the description of the chosen statistical analysis methods in our future work.
>
>
>  W2: Evaluation metrics like ACC-IN and ACC-IN-C are not explained.
>
>  A2: ACC-IN and ACC-IN-C are commonly used accuracy for the two test sets, i.e., in-distribution data (Non-IID data) and covariate-shift data (OOD Generalization), respectively. We follow FedTHE to choose ACC-IN and ACC-IN-C. We will explain it in detail in our future version.
>
>
>  W3: The related work section only introduces some comparison methods and lacks more related work, such as FedAvg. Besides, some references are incomplete, such as ref. 46.
>
> A3: We provide the illustration of FedAvg in line 671 of supplemental pages, and ref. 46 is the SVHN dataset for detecting OUT samples. We will elaborate the related work during reversion.
>
>
> W4: No open-source code has been provided.
>
> A4: We will release our project in the later version, and we have released the test model trained on Cifar10 $\alpha=0.5$ for FedAvg-FOOGD, in our supplements, for validating the effectiveness of FOOGD.
>
>  Q1: Could the experimental results section provide a more detailed analysis of OOD to better explain the superiority of the FOOGD method in this aspect?
>
> A5: **Yes, we can provide more detailed analysis and corresponding evaluation intention in our future vesion.**
> FOOGD demonstrates its superiority in OOD shifts data by three main aspects:
> (1) In Tab. 1-2 and Tab. 4-5: Superior performance of non-IID data classification, OOD detection and generalization for setup of different degrees of non-IID, and various datasets (Cifar10, Cifar100, TinyImageNet, PACS).
> (2) In Fig. 4:  FOOGD has more separable clustering among IN, IN-C, and OUT data, in feature space, and FOOGD pushes OUT data away from its IN and IN-C data, which validates the guidance from the global distribution.
> (3) In Fig.6 (Tab. 6) and Fig.7: Extensive OOD detection  and generalization tasks consistently achieve performance enhancement, affirming the reliable performance generalization of FPPGD in handling OOD data.
> We will improve the the details of relavant empirical studies, in our future version.

---

> > ### Comment · Reviewer_cFix · 2024-08-10
> >
> > Thank you for the response. However, I am unhappy with the response since it did not address my concerns completely. All the problems are planned to be addressed in future work rather than the current version, and the authors did not provide evidence to address the issues in the response. Thus, I will make a weak reject decision for the paper.

---

> ### Author Response · Authors · 2024-08-10
> **Response to Reviwer cFix**
>
> **We are not planning to address the problems you raised in future work, instead, we reverse these problems in our current work. The future version indicates that the camera-ready version we revised based on the current version, because it is impossible to resubmit on the current version.**
> We elaborate our revised version in the following.
>
> **The evidential results for weakness 1 is explained in our A1, where we provide the reason for choosing accuracy for OOD generalization, and FPR95 and AUROC for OOD detection, which statistically follow the work of FedTHE, FOSTER, and SCONE.** This is introduced in lines 272-282 of main paper, and lines 654-663 in appendix. We omit the statistics in our rebuttal due to space limits. And we provide the revised two main tables as below.
>
> **Table 1: Main results of federated OOD detection and generalization on Cifar10. We report the ACC of brightness as IN-C ACC, the FPR95 and AUROC of LSUN-C as OUT performance.**
> | Non-IID\Method | $\alpha=0.1$ |  |  |  | $\alpha=0.5$ |  |  |  | $\alpha=5$  |  |  |  |
> | :---: | :---: | :---: | :---: | :---: | :---: | :---: | :---: | :---: | :---: | :---: | :---: | :---: |
> |  | ACC-IN $\uparrow$ | ACC-IN-C $\uparrow$ | FPR95 $\downarrow$ | AUROC $\uparrow$ | ACC-IN $\uparrow$ | ACC-IN-C $\uparrow$ | FPR95 $\downarrow$ | AUROC $\uparrow$ | ACC-IN $\uparrow$ | ACC-IN-C $\uparrow$ | FPR95 $\downarrow$ | AUROC $\uparrow$ |
> | FedAvg | 68.03 $\pm$ 1.17 |    65.44 $\pm$ 1.18   | 83.41 $\pm$ 1.57 | 58.05 $\pm$ 0.89 | 86.59 $\pm$ 1.13 |    83.72 $\pm$ 1.74   | 43.70 $\pm$ 0.83 | 84.18 $\pm$ 0.23 | 86.50 $\pm$ 0.33 |    85.08 $\pm$ 0.49   | 38.24 $\pm$ 0.55 | 85.37 $\pm$ 0.29 |
> | FedLN | 75.24 $\pm$ 0.44 |    71.77 $\pm$ 0.67   | 56.14 $\pm$ 0.91 | 84.14 $\pm$ 0.37 | 86.10 $\pm$ 0.89 |    84.20 $\pm$ 1.82   | 39.26 $\pm$ 1.14 | 89.64 $\pm$ 0.52 | 87.20 $\pm$ 1.26 |    85.08 $\pm$ 1.43   | 33.33 $\pm$ 2.38 | 90.87 $\pm$ 0.58 |
> | FedATOL | 55.93 $\pm$ 1.87 |    54.44 $\pm$ 1.72   | 49.50 $\pm$ 1.59 | 86.22 $\pm$ 2.74 | 87.55 $\pm$ 0.91 |    85.64 $\pm$ 0.54   | 27.87 $\pm$ 1.32 | 93.48 $\pm$ 0.69 | 89.27 $\pm$ 0.68 |    88.28 $\pm$ 1.32   | 19.66 $\pm$ 2.62 | 95.25 $\pm$ 0.78 |
> | FedT3A | 68.03 $\pm$ 1.17 |    61.52 $\pm$ 1.39   | 83.41 $\pm$ 1.57 | 58.05 $\pm$ 0.89 | 86.59 $\pm$ 1.13 |    82.85 $\pm$ 0.44   | 43.70 $\pm$ 0.83 | 84.18 $\pm$ 0.23 | 86.50 $\pm$ 0.33 |    85.01 $\pm$ 1.46   | 38.24 $\pm$ 0.55 | 85.37 $\pm$ 0.29 |
> | FedIIR | 68.26 $\pm$ 0.66 |    66.12 $\pm$ 0.74   | 79.48 $\pm$ 0.99 | 63.31 $\pm$ 1.38 | 86.75 $\pm$ 0.98 |    84.75 $\pm$ 1.92   | 40.91 $\pm$ 0.64 | 84.94 $\pm$ 0.49 | 87.77 $\pm$ 0.66 |    86.10 $\pm$ 0.95   | 34.69 $\pm$ 1.07 | 87.66 $\pm$ 0.47 |
> | FedAvg+FOOGD | 75.09 $\pm$ 0.79 |    73.71 $\pm$ 0.93   | 35.32 $\pm$ 1.02 | 91.21 $\pm$ 0.78 | 88.36 $\pm$ 0.43 |    87.26 $\pm$ 0.86   | 17.78 $\pm$ 0.62 | 96.53 $\pm$ 0.18 | 88.90 $\pm$ 0.29 |    88.25 $\pm$ 0.12   | 12.02 $\pm$ 0.34 | 97.77 $\pm$ 0.41 |
> | FedRoD | 91.15 $\pm$ 0.87 |    89.90 $\pm$ 0.85   | 47.97 $\pm$ 1.88 | 80.96 $\pm$ 0.90 | 89.62 $\pm$ 0.55 |    87.70 $\pm$ 0.80   | 37.03 $\pm$ 1.40 | 86.50 $\pm$ 0.97 | 87.69 $\pm$ 0.88 |    86.26 $\pm$ 1.19   | 36.13 $\pm$ 1.12 | 86.65 $\pm$ 0.36 |
> | FOSTER | 90.22 $\pm$ 0.88 |    88.70 $\pm$ 0.82   | 47.40 $\pm$ 1.27 | 77.44 $\pm$ 0.93 | 86.92 $\pm$ 1.85 |    85.82 $\pm$ 1.10   | 42.03 $\pm$ 1.51 | 83.91 $\pm$ 1.11 | 87.83 $\pm$ 1.38 |    85.96 $\pm$ 1.02   | 36.42 $\pm$ 1.14 | 86.19 $\pm$ 0.87 |
> | FedTHE | 91.05 $\pm$ 0.66 |    89.77 $\pm$ 0.91   | 58.14 $\pm$ 2.79 | 82.04 $\pm$ 1.15 | 89.14 $\pm$ 0.93 |    87.68 $\pm$ 0.41   | 40.28 $\pm$ 2.43 | 85.30 $\pm$ 1.91 | 88.14 $\pm$ 0.24 |    86.18 $\pm$ 0.57   | 35.35 $\pm$ 1.94 | 86.79 $\pm$ 0.37 |
> | FedICON | 89.06 $\pm$ 0.43 |    89.18 $\pm$ 0.81   | 48.22 $\pm$ 1.48 | 81.28 $\pm$ 0.44 | 75.83 $\pm$ 1.07 |    75.35 $\pm$ 0.36    | 56.19 $\pm$ 1.58 | 79.88 $\pm$ 0.51 | 87.20 $\pm$ 1.13 |    85.39 $\pm$ 0.99   | 35.63 $\pm$ 1.16 | 86.45 $\pm$ 0.41 |
> | FedRoD+FOOGD | 93.51 $\pm$ 0.65 |    92.74 $\pm$ 0.46   | 32.99 $\pm$ 1.30 | 91.76 $\pm$ 0.26 | 90.46 $\pm$ 0.78 |    90.16 $\pm$ 0.51   | 25.51 $\pm$ 1.46 | 94.19 $\pm$ 0.78 | 89.44 $\pm$ 0.88 |    88.62 $\pm$ 0.37   | 18.91 $\pm$ 0.46 | 96.25 $\pm$ 0.22 |

---

> ### Author Response · Authors · 2024-08-10
> **Response to Reviewer cFix**
>
> Table 2: Main results of federated OOD detection and generalization on Cifar100. We report the ACC of brightness as IN-C ACC, the FPR95 and AUROC of LSUN-C as OUT performance.
>
> | Non-IID \ Method | $\alpha=0.1$ |  |  |  | $\alpha=0.5$ |  |  |  | $\alpha=5$  |  |  |  |
> | :---: | :---: | :---: | :---: | :---: | :---: | :---: | :---: | :---: | :---: | :---: | :---: | :---: |
> |  | ACC-IN $\uparrow$ | ACC-IN-C $\uparrow$ | FPR95 $\downarrow$ | AUROC $\uparrow$ | ACC-IN $\uparrow$ | ACC-IN-C $\uparrow$ | FPR95 $\downarrow$ | AUROC $\uparrow$ | ACC-IN $\uparrow$ | ACC-IN-C $\uparrow$ | FPR95 $\downarrow$ | AUROC $\uparrow$ |
> | FedAvg | 51.67 $\pm$ 1.37 |    47.54 $\pm$ 0.48   | 78.35 $\pm$ 1.64 | 67.16 $\pm$ 1.17 | 58.28 $\pm$ 0.48 |    54.62 $\pm$ 0.67   | 72.84 $\pm$ 0.81 | 70.86 $\pm$ 1.52 | 61.40 $\pm$ 0.12 |    56.72 $\pm$ 0.17   | 72.68 $\pm$ 0.34 | 70.59 $\pm$ 0.19 |
> | FedLN | 52.48 $\pm$ 1.41 |    48.15 $\pm$ 1.57   | 66.94 $\pm$ 1.61 | 74.82 $\pm$ 0.50 | 59.39 $\pm$ 0.72 |    53.86 $\pm$ 1.23   | 68.31 $\pm$ 1.24 | 73.41 $\pm$ 0.33 | 61.00 $\pm$ 0.40 |    56.33 $\pm$ 0.82   | 69.18 $\pm$ 0.46 | 75.87 $\pm$ 0.74 |
> | FedATOL | 43.65 $\pm$ 0.54 |    41.08  $\pm$  0.60   | 65.26 $\pm$ 0.96 | 81.64 $\pm$ 0.33 | 60.62 $\pm$ 0.61 |    56.63 $\pm$ 0.91   | 70.10 $\pm$ 0.81 | 79.27 $\pm$ 0.61 | 64.16 $\pm$ 0.81 |    63.61 $\pm$ 0.42   | 80.27 $\pm$ 1.61 | 60.51 $\pm$ 1.75 |
> | FedT3A | 51.67 $\pm$ 1.37 |    51.50 $\pm$ 0.29   | 78.35 $\pm$ 1.64 | 67.16 $\pm$ 1.17 | 58.28 $\pm$ 0.48 |    55.42 $\pm$ 1.63   | 72.84 $\pm$ 1.56 | 70.86 $\pm$ 1.52 | 61.40 $\pm$ 0.12 |    55.51 $\pm$ 0.96   | 72.68 $\pm$ 0.34 | 70.59 $\pm$ 0.19 |
> | FedIIR | 51.63 $\pm$ 0.61 |  47.88 $\pm$ 1.19   | 81.91 $\pm$ 0.47 | 63.99 $\pm$ 0.53 | 58.66 $\pm$ 0.41 |    55.72 $\pm$ 0.29   | 77.62 $\pm$ 1.10 | 65.87 $\pm$ 0.46 | 61.70 $\pm$ 0.76 |    57.65 $\pm$ 0.80   | 72.57 $\pm$ 0.37 | 69.07 $\pm$ 0.52 |
> | FedAvg+FOOGD | 53.84 $\pm$ 0.83 |    51.69 $\pm$ 0.32   | 36.40 $\pm$ 1.11 | 91.41 $\pm$ 0.36 | 61.82 $\pm$ 0.20 |    59.91 $\pm$ 0.31   | 55.70 $\pm$ 0.78 | 86.42 $\pm$ 0.24 | 64.96 $\pm$ 0.51 |    64.18 $\pm$ 0.31   | 57.70 $\pm$ 0.87 | 84.03 $\pm$ 0.15 |
> | FedRoD | 73.13 $\pm$ 0.85|69.26 $\pm$ 0.41| 66.34 $\pm$ 1.53 | 73.02 $\pm$ 1.82 | 66.88 $\pm$ 0.61 |    61.28 $\pm$ 0.98   | 70.13 $\pm$ 0.86 | 69.48 $\pm$ 0.65 | 61.34 $\pm$ 0.78 |    55.80 $\pm$ 1.21   | 74.86 $\pm$ 0.98 | 67.76 $\pm$ 1.31 |
> | FOSTER | 72.54 $\pm$ 1.51 |    67.50 $\pm$ 0.57| 61.25 $\pm$ 1.05 | 75.44 $\pm$ 0.89 | 62.45 $\pm$ 0.55 |    57.62 $\pm$ 0.87   | 73.26 $\pm$ 1.13 | 68.71 $\pm$ 0.85 | 53.80 $\pm$ 0.31 |    49.28 $\pm$ 0.74   | 76.94 $\pm$ 1.62 | 65.47 $\pm$ 1.72 |
> | FedTHE | 73.83 $\pm$ 0.48 |    69.09 $\pm$ 0.56   | 64.73 $\pm$ 0.79 | 75.16 $\pm$ 0.34 | 66.22 $\pm$ 0.68 |61.19 $\pm$ 0.92   | 72.95 $\pm$ 1.84 | 69.38 $\pm$ 1.64 | 61.03 $\pm$ 0.22 |    57.03 $\pm$ 0.16   | 71.43 $\pm$ 0.64 | 69.01 $\pm$ 0.87 |
> | FedICON | 72.22 $\pm$ 0.72 |67.79 $\pm$ 0.31   | 61.36 $\pm$ 0.39 | 77.12 $\pm$ 0.55 | 65.86 $\pm$ 0.81 |61.83 $\pm$ 0.55   | 69.99 $\pm$ 1.02 | 71.03 $\pm$ 0.39 | 62.11 $\pm$ 0.74|57.62 $\pm$ 0.28   | 70.91 $\pm$ 0.97 | 70.84 $\pm$ 0.73 |
> | FedRoD+FOOGD | 77.88 $\pm$ 0.28 |75.70 $\pm$ 0.26|58.81 $\pm$ 0.48 | 86.07 $\pm$ 0.39 |  70.30 $\pm$ 0.46 |    68.23 $\pm$ 0.25| 45.19 $\pm$ 0.67 | 89.59 $\pm$ 0.28 | 64.94 $\pm$ 0.79 |    62.56 $\pm$ 0.72   | 65.18 $\pm$ 1.19 |80.47 $\pm$ 0.32|
>
> **As we can see, the model performance variance of FOOGD is very small, indicating the performance stability of FOOGD in OOD data. This also validates that our model is effective and practical for federated learning with wild data.**
>
> **For weakness 2, we reply that ACC-IN and ACC-IN-C are commonly used accuracies for the two test sets, i.e., in-distribution data (Non-IID data) and covariate-shift data (OOD Generalization), respectively**. We have introduced them in lines 278-2799 of the main paper. This is used in federated learning considering semantic shifts, e.g., FedTHE, where ACC-IN corresponds to original local test, and ACC-IN-C corresponds to corrupted local test.
>
> **For weakness 3, the related work, e.g., FedAvg as well as their implementations are listed in lines 664-693, please kindly refer to this part for more details.** In this work, we mainly focus on tackle the OOD shifts in federated learning, which is orthogonal to the existing work that tackles heterogeneity. We will add a section for introducing work related on FL with non-IID data.
> **The reference you indicated, i.e., ref.46, is the SVHN dataset that is commonly used in detecting OUT samples.** SVHN is a real-world image dataset from Google Street View house numbers, comprising 73,257 training samples and 26,032 testing samples across 10 classes. We introduce it as an OOD detection task in line 267  and line 646 of paper.

---

> ### Author Response · Authors · 2024-08-10
> **Response to Reviewer cFix**
>
> **For weakness 4, we release our project in an anonymous page: https://anonymous.4open.science/r/FOOGD/, and we have released the test model trained on Cifar10 $\alpha=0.5$ for FedAvg-FOOGD, in our supplements, for validating the effectiveness of FOOGD.**
>
> **For the response of question 1, please point out your concerns directly. We are willing to clarify all the misunderstandings.**
>
> In summary, we believe that some previous rebuttals might have caused a misunderstanding. We hope you can reconsider our response to see if it resolves your issue. If you have any other questions, please kindly consider responding; we would be happy to discuss them. Please provide us your additional concens as soon as possible, since the the rebuttal time is approaching the end.

---

> ### Comment · Reviewer_cFix · 2024-08-11
>
> Thank you for the detailed response. Some concerns have been addressed in the response, so I have raised the mark to the positive rating.

---

> > ### Author Response · Authors · 2024-08-11
> > **Response to reviwer cFix**
> >
> > Thank you for your timely response and insightful reviwes. We are delighted to have addressed your concens and to know that you appreciate our work! The conversation with you are greatly beneficial for our work, such as providing mor detailed explainations. Once again, we are sincerely gratitude for your thorough and patient reviews.

---

### Official Review · Reviewer_MUVw · 2024-07-09

**Soundness:** 3
**Presentation:** 3
**Contribution:** 3
**Rating:** 7
**Confidence:** 4

**Summary:**

This paper addresses various OOD (Out-of-Distribution) shifts that may occur in federated settings by proposing a unified framework to simultaneously tackle OOD generalization and detection issues. Specifically, this paper introduces FOOGD, which estimates arbitrary client probability densities to create a reliable global distribution that guides both OOD generalization and detection. Besides, it presents $SM^3D$ and SAG to respectively enhance OOD detection and generalization capabilities. Finally, extensive experiments demonstrate the effectiveness of the proposed methods

**Strengths:**

1. This paper proposes a novel unified framework to simultaneously address OOD (Out-of-Distribution) generalization and detection problems in federated settings.

2. The method has theoretical guarantees and empirical analysis to ensure its effectiveness.

3. The structure of this work is logical, and the writing is well-organized.

**Weaknesses:**

1. This paper proposes the FOOGD, $SM^3D$, and SAG methods to address OOD generalization and detection problems. It is essential to clearly define the relationships among these three methods in the abstract and introduction to enhance the readability of the entire paper.

2. This paper uses 'semantic-shift' to describe the issue of unknown class bias in OOD detection, following the traditional machine learning concept[1]. However, in federated learning, this problem is generally referred to as 'concept shift'[2][3]. It is recommended to revise this terminology to align better with the conventions of the federated learning community.

3. In the related work section (Section 2.3), it is recommended to include a discussion on 'concept shift' in federated learning and cite relevant literature.

4. In the methodology section, given the involvement of multiple innovative components, it is recommended to analyze their complexity.


References:

[1] Haoyue Bai et al. Feed two birds with one scone: Exploiting wild data for both out-of-distribution generalization and detection. ICML, 2023.

[2] Panchal K et al. Flash: concept drift adaptation in federated learning. ICML, 2023.

[3] Jothimurugesan E et al. Federated learning under distributed concept drift. ICAI, 2023.

**Questions:**

See weaknesses.

**Limitations:**

See weaknesses.

---

> ### Author Rebuttal · Authors · 2024-08-04
>
> Thanks for your constructive review! For your concerns, we explain them in the following.
>
> W1:  It is essential to clearly define the relationships among FOOGD, $SM^3D$, and SAG, in the abstract and introduction to enhance the readability of the entire paper.
>
>  A1: FOOGD is the overall federated learning framework for adapting wild data.
>  In FOOGD, each client or server maintains a feature extractor and a score model.
>  And $SM^3D$ is devised for training a score model that estimates distribution and detects OUT data.
>  While SAG is proposed to enhance feature extractor in the main task, e.g., classification, as well as generalization based on score estimation.
>  We will clarify it in the abstract and introduction of our later version.
>
>  W2: However, in federated learning, this problem is generally referred to as 'concept shift'[2][3]. It is recommended to revise this terminology to align better with the conventions of the federated learning community.
>
> A2: **FOOGD is proposed for adapting non-IID data and OOD shifts data, which is orthogonal with FL methods solving 'concept shift'.** OOD shifts are commonly existing in federated scenarios, and various FL methods are proposed for OOD detection, e.g., FOSTER, and OOD generalization, e.g., FeDIIR and FedTHE.
>  **It is possible for FOOGD to resolve the 'concept shift' issue, similarly following the effectiveness in OOD generalization.**
>  Specifically, FOOGD adaptively estimates the arbitrary data distribution, and aggregates the global distribution on the server. Then FOOGD utilizes the newly updated score model as distribution guidance, for concept shift data.
>  Since FOOGD utilizes global distribution and outperforms on the tasks of new unseen clients with PACS data, it can adapt to concept shift data as well. We are willing to elaborate the relationship between FL with OOD data and 'concept shift' data in our later version.
>
>
>
>  W3: In the related work section (Section 2.3), it is recommended to include a discussion on 'concept shift' in federated learning and cite relevant literature.
>
>  A3: We are willing to extend our related work discussion on 'concept shift' in federated learning and cite relevant literature, e.g., Flash and FedDrift.
>
>
>
>  W4: In the methodology section, given the involvement of multiple innovative components, it is recommended to analyze their complexity.
>
>
>  A4: For newly added modules, the complexity of $SM^3D$ is $O((T+1)BD + B^2 D)$, and the complexity of SAG is $O(B^2 D)$, where $B$ is the batch size of samples, $T$ is the steps of sampling, and $D$ is the feature dimension .
>
> **$SM^3D$ is responsible for estimating the score model for density estimation and OOD detection. The main procedure of $SM^3D$ consists of score matching (SM), Langevin Dynamic sampling (LDS), and Maximum Mean Discrepancy (MMD) computation.**
>
> - **Score Matching:** The complexity of computing score matching is $O(B D)$.
> - **Langevin Dynamic Sampling (LDS):** This involves $T$ steps of sampling, each with a complexity of $O(B D)$. Therefore, the overall complexity for LDS is $O(T B D)$.
> - **Maximum Mean Discrepancy (MMD):** Computing the MMD involves pairwise comparisons of samples, leading to a complexity of $O(B^2 D)$.
>
> Combining these, the complexity of $SM^3D$ is $O((T+1)BD +  B^2 D)$.
>
> **SAG enhances the generalization capability of the feature extractor by aligning distributions of original and augmented data measured by Kernelized Stein Discrepancy (KSD).**
>
> - **Kernelized Stein Discrepancy (KSD):** This involves computing gradients and kernel functions between pairs of features. The complexity is $O(B^2 D)$, where $B$ is the batch size of samples, and $D$ is the feature dimension.
>
> Thus, the complexity of SAG is $O(B^2 D)$.
>
>  We also provide the empirical reports in Tab. a-b. It is obvious that the most computation cost stems from main task inference. Though the newly modules bring additional cost, the overall burden is controllable and effective in OOD data.
>
>   **Table a. Average computation cost of batch samples for training score model.**
> |            	  	| Main task | $SM^3D$ | Total |
> |--------------------|-------------------|-----|-----|
> | Cifar10 $\alpha=0.1$ (s)  | 0.0252	   | 0.0031  | 0.0283 |
> | cifar100 $\alpha=0.1$ (s) | 0.0281    	| 0.0031   | 0.0312   |
>  ||
>
>
>   **Table b. Average computation cost of batch samples for training feature extractor.**
> |        	  	 	| Main task | SAG | Total |
> |--------------------|-------------------|-----|-----|
> | Cifar10 $\alpha=0.1$ (s)  | 0.0503      | 0.00212  | 0.0524 |
> | cifar100 $\alpha=0.1$ (s) | 0.0561		| 0.00236   | 0.0585   |
>  ||

---

> > ### Comment · Reviewer_MUVw · 2024-08-09
> >
> > It clearly addresses my concerns, and I will raise my score.

---

> > > ### Author Response · Authors · 2024-08-11
> > > **Response to Reviewer MUVw**
> > >
> > > We are pleased that our work has received professional evaluation and recognition. We are more than willing to incorporate the suggested revisions into our camera-ready version to address the issues raised.

---

### Official Review · Reviewer_RpTd · 2024-07-13

**Soundness:** 4
**Presentation:** 4
**Contribution:** 3
**Rating:** 8
**Confidence:** 4

**Summary:**

This paper focuses on the federated learning setup that non-IID, semantic-shift, and covariate-shift take place in the same time. And the authors propose a framework of FOOGD with a unnormalized distribution estimation method, i.e., SM3D, to release the distribution assumption and constraints in heterogeneous modeling. Both federated OOD generalization and detection can be soundly solved via SM3D and SAG modules. Compared with existing methods, FOOGD is validated via extensive experiments for effectiveness.

After rebuttal, I raise the Rating from 7 to 8.

**Strengths:**

1.	The presentation is straightforward and clear, with detailed and instructive figures. The formula derivation is thorough and essential.
2.	The paper addresses a significant challenge in federated learning, focusing on learning from large-scale non-IID data with out-of-distribution (OOD) shifts.
3.	The authors conduct extensive and robust experiments, effectively demonstrating the effectiveness of the proposed FOOGD method.
4.	Detailed theoretical foundations are laid out for both the estimation model and the consistency of convergence optimization.
5.	Utilizing score modeling for federated learning is quite novel and insightful for heterogeneously distributed data.

**Weaknesses:**

1.	How to understand IsOUT() function with negative threshold in Equation 9?
2.	Score model brings additional computation cost.
3.	The correspondence between federated loss of $\mathcal{L}_k$ and the practical loss $\mathcal{L}_s$ and $\mathcal{L}_f$ are not explicitly explained.

**Questions:**

1.	How to enhance the generalization for unseen client distribution?

**Limitations:**

Yes

---

> ### Author Rebuttal · Authors · 2024-08-04
>
> Thank you for appreciating the contributions of our work. For the weakness concerned by the reviewers, we provide the discussion as below.
>
> W1: How to understand IsOUT() function with negative threshold in Eq. (9)?
>
>  A1:IsOUT() function is the OOD detection function in our work. When the norm of score function of the data sample is larger than a given threshold, the probability of OUT is higher. The threshold $\tau$ can be both positive values and negative values.
>  If the threshold $\tau>0$, the bigger the norm than $\tau$ indicates more potential to be OUT data samples.
>
> $\operatorname{IsOUT}(\boldsymbol{x})=$ True, when $\left\|s_{\boldsymbol{\theta}^*}\left(f_{\boldsymbol{\theta}^*}(\boldsymbol{x})\right)\right\|>\tau ; \quad$ otherwise, IsOUT $(\boldsymbol{x})=$ False.
>
>  On the contrary, if the threshold $\tau<0$, the bigger the norm than $-\tau$ indicates more potential to be OUT data:
>
> $\operatorname{IsOUT}(\boldsymbol{x})=$ True, when $\left\|s_{\boldsymbol{\theta}^*}\left(f_{\boldsymbol{\theta}^*}(\boldsymbol{x})\right)\right\|>-\tau ; \quad$ otherwise, IsOUT $(\boldsymbol{x})=$ False.
>
> Both formulations of OUT detection functions are the same, we choose the later one in order to have approximately consistent visualization with existing detection scores, e.g., MSP, in our experiments.
> That is, in Fig. 5 of main paper, the detection score distributions are consistently in the order, i.e., OUT in the left, IN-C in the middle, and IN in the right.
>
>
>  W2: Score model brings additional computation cost.
>
> A2: **The computation cost of score model is slightly increasing but controllable.**
>  Specifically, we estimate the score model in latent representation space, which reduces the computation cost of score function of original image data[1]. Besides, in implementation, we choose small 3-layer MLP models to capture the distribution in 512-dimentional latent space.
>  In Tab. a, we provide the computing cost of different modules in FOOGD, e.g., forward of main task, score estimation of $SM^3D$. The most computation burden is due to main task inference, i.e., 88.97%-90.00% of the total computation time.
>  Though computation is added for modeling distribution, FOOGD enhances its robustness for OOD shifts data. We believe the advantages of FOOGD outperforms its short-comings.
>
>   **Table a. Average computation cost of batch samples for training score model.**
> |            	  	| Main task | $SM^3D$ | Total |
> |--------------------|-------------------|-----|-----|
> | Cifar10 $\alpha=0.1$ (s)  | 0.0252	   | 0.0031  | 0.0283 |
> | cifar100 $\alpha=0.1$ (s) | 0.0281    	| 0.0031   | 0.0312   |
>  ||
>
>
> [1] Vahdat A, Kreis K, Kautz J. Score-based generative modeling in latent space[J]. Advances in neural information processing systems, 2021, 34: 11287-11302.
>
>
>  W3: The correspondence between federated loss of $\mathcal{L}_k$ and the practical loss $\mathcal{L}_s$ and $\mathcal{L}_f$ are not explicitly explained.
>
>  A3: The loss $\mathcal{L}_k$ is the general optimization objective in client $k$, while $\mathcal{L}_s$ enhance the score estimation for detection which corresponds with the implementation of $\ell_k^{OUT}$, besides, $\mathcal{L}_f$ enhance feature extractor for generalization, which further corresponds to $\ell_k^{IN}$ and $\ell_k^{IN-C}$.
>  In summary,
> $\mathcal{L}_k= \ell_k^{IN}+ \ell_k^{IN-C}+ \ell_k^{OUT}= \mathcal{L}_s+\mathcal{L}_f$.
> We will clarify this relationship in our later version.
>
> Q1: How to enhance the generalization for unseen client distribution?
>
> A: In FOOGD, we maintain a score model that estimates the overall global distribution of in-distribution data. Therefore, for any new client, we can warm-start its local score model with a global one, which is directly applicable. In Table 4 of our empirical study main paper, we evaluate FOOGD on PACS by leaving one domain untrained, achieving competitive performance. This validates the effectiveness of unseen clients.

---

> > ### Comment · Reviewer_RpTd · 2024-08-08
> > **Thanks for your response**
> >
> > Thank you for the detailed response, the author effectively addressed my issue, I will increase my grade.
> > Hope you can add these details to your appendix.

---

> > > ### Author Response · Authors · 2024-08-11
> > > **Response to reviewer RpTd**
> > >
> > > Thank you for appreciating our work. We will make sure to include the relevant details and the addressed specific issues in the camera-ready version.

---

### Decision · Program_Chairs · 2024-09-25

**Decision:**

Accept (poster)

**Comment:**

This paper focuses on simultaneously dealing with OOD detection and OOD generalization in FL scenarios for the first time. To solve this novel challenge, the proposed method, FOOGD, that estimates the probability density of each client to derive the global data distribution using SM3D, and employs SAG to train a feature extractor that captures invariant yet diverse knowledge, enabling both local covariate shift generalization and inter-client generalization. Extensive experiments are conducted to demonstrate the effectiveness of the proposed method.
The reviewers initially suggested the following strong points and concerns.

Strong points:
- FOOGD is the first framework in federated learning to simultaneously address both OOD generalization and detection, effectively tackling a significant challenge in the field.

- The use of score modeling in federated learning is both novel and insightful, particularly for managing heterogeneously distributed data.

- The theoretical and empirical analyses effectively demonstrate the effectiveness of the proposed method.

Concerns:

- The computational costs of the proposed method seem to be high (as noted by all reviewers).

- Certain sections lack sufficient explanations or detailed content.

- There are privacy concerns regarding the transmission of detailed model parameters and score function values between clients and the server.

- The effectiveness of the proposed method appears to heavily depend on the accuracy of the score modeling, raising concerns in scenarios where data distributions are highly complex or sparse.

The authors have adequately addressed the reviewers' concerns, leading all reviewers to raise their initial scores. The authors are advised to incorporate these changes into the final version of the paper.